# A TRAF-like E3 ubiquitin ligase TrafE coordinates ESCRT and autophagy in endolysosomal damage response and cell-autonomous immunity to *Mycobacterium marinum*

**Lyudmil Raykov, Manon Mottet, Jahn Nitschke, Thierry Soldati***

Départment de Biochimie, Faculté des Sciences, University of Geneva, Geneva, Switzerland

**\*For correspondence:**
thierry.soldati@unige.ch

**Competing interest:** The authors declare that no competing interests exist.

**Abstract** Cells are perpetually challenged by pathogens, protein aggregates or chemicals, that induce plasma membrane or endolysosomal compartments damage. This severe stress is recognised and controlled by the endosomal sorting complex required for transport (ESCRT) and the autophagy machineries, which are recruited to damaged membranes to either repair or to remove membrane remnants. Yet, insight is limited about how damage is sensed and which effectors lead to extensive tagging of the damaged organelles with signals, such as K63-polyubiquitin, required for the recruitment of membrane repair or removal machineries. To explore the key factors responsible for detection and marking of damaged compartments, we use the professional phagocyte *Dictyostelium discoideum*. We found an evolutionary conserved E3-ligase, TrafE, that is robustly recruited to intracellular compartments disrupted after infection with *Mycobacterium marinum* or after sterile damage caused by chemical compounds. TrafE acts at the intersection of ESCRT and autophagy pathways and plays a key role in functional recruitment of the ESCRT subunits ALIX, Vps32 and Vps4 to damage sites. Importantly, we show that the absence of TrafE severely compromises the xenophagy restriction of mycobacteria as well as ESCRT-mediated and autophagy-mediated endolysosomal membrane damage repair, resulting in early cell death.

## Editor's evaluation

This study presents important findings on the mechanism as to how *Mycobacterium*-containing vacuoles are recognized by host cell factors and subjected to membrane repairment or autophagic degradation using *Dictyostelium discoideum* as a useful model. The evidence for the role of TrafE in damaged membrane repair and xenophagy induction is convincing. This work will be of interest to cell biologists and microbiologists.

## Introduction

After its internalisation by professional phagocytes, the first strategy of *Mycobacterium tuberculosis* and its close relative *Mycobacterium marinum* to escape the bactericidal phagosome is to impede its maturation and tailor the *Mycobacterium*-Containing Vacuole (MCV) as a permissive niche allowing the proliferation of the pathogen (*Soldati and Neyrolles, 2012*). From the very beginning, bacteria progressively gain access to the cytosol through the action of the ESX-1 secretion system, via the secretion of the membranolytic effector EsxA and the mycobacterial branched apolar lipids phthiocerol

dimycocerosates (PDIMs)( *Pym et al., 2002*; *Lerner et al., 2018*; *Quigley et al., 2017*; *Augenstreich et al., 2019*; *Lienard et al., 2020*; *Osman et al., 2020*). The ESX-1 secretion system plays a crucial role for *M. marinum* virulence and intracellular growth, and is primarily encoded by the genes within the 'region of difference 1' (RD1) locus. *M. marinum* mutants with non-functional ESX-1 inflict significantly less MCV membrane damage and fail to become exposed to or escape to the cytosol, hampering their detection by the host cell-autonomous defence machineries (*Cardenal-Muñoz et al., 2017*; *Gao et al., 2004*). Once in contact with the cytosol, the wild-type (WT) bacteria within damaged MCVs are detected and tagged with signals that promote MCV damage repair as well as selective anti-bacterial autophagy, termed xenophagy (*López-Jiménez et al., 2018*). To complete the infection cycle, the pathogen must reach the cytosol and be released by host-cell lysis, exocytosis or ejection in order to disseminate to adjacent bystander cells (*Hagedorn et al., 2009*; *Queval et al., 2017*; *Cardenal-Muñoz et al., 2017*; *López-Jiménez et al., 2018*).

Xenophagy is fundamental for eukaryotic cell-autonomous immunity (*Sharma et al., 2018*; *Deretic et al., 2013*). It requires the coordinated action of about 15 autophagy-related genes (ATG) and is initiated by the recruitment of the core-autophagy factors Atg1 (ULK1 in mammals) kinase complex and Atg9 (ATG9 in mammals) to the nascent phagophore – a double membrane crescent-like structure. Its closure around the cargo depends on the Atg16-Atg12-Atg5 E3-like complex, and the lipidation of Atg8 (LC3 in mammals), which marks the phagophore (*Mesquita et al., 2017*). The resulting autophagosome fuses with lysosomes allowing the degradation of its content. Xenophagy requires detection and selective recruitment of the pathogen to the phagophore by means of autophagy cargo receptors such as p62 (SQSTM1 in mammals) that link ubiquitin-tagged bacteria and host components with Atg8 family proteins on the phagophore membranes, thereby enforcing proximity between the pathogen and the autophagosome (*Boyle and Randow, 2013*). Autophagy is involved in the removal of extensively damaged compartments by a specific macroautophagy termed lysophagy. Autophagy also cooperates with ESCRT to repair sterile- or pathogen-induced membrane damage (*Skowyra et al., 2018*; *López-Jiménez et al., 2018*; *Papadopoulos et al., 2020*). The ESCRT-III complex functions in membrane deformation away from the cytosol and plays a critical role in various membrane remodelling processes, including phagophore closure, autophagosome-lysosome fusion, plasma membrane, and lysosome membrane repair (*Vietri et al., 2020*; *Schuck, 2020*). One thoroughly studied ESCRT function is the coupling of cargo sorting and intralumenal vesicles formation required for degradation of receptors.

The endolysosomal and MCV membrane ruptures generate morphological and topological landscape for the detection machineries which deposit 'repair-me' and 'eat-me' signals (*Boyle and Randow, 2013*; *Koerver et al., 2019*; *Papadopoulos et al., 2020*). Ubiquitination is a well-studied, highly conserved and versatile post-translational modification used as a signalling platform in diverse catabolic processes and playing a role during infection with various intracellular bacteria. In mammalian cells, the RING domain E3 ligases LRSAM1 and TRAF6 have been shown to promote ubiquitination of vacuoles containing *Salmonella* Typhimurium and *Chlamydia trachomatis* bacteria, and the protozoan parasite *Toxoplasma gondii* (*Huett et al., 2012*; *Haldar et al., 2015*), and NEDD4 (*Pei et al., 2017*), Parkin (*Manzanillo et al., 2013*), Smurf1 (*Franco et al., 2017*) to mediate ubiquitination of mycobacteria and host proteins during infection. In addition, the mammalian cytosolic lectins, galectins, recognize damage by binding exposed glycans on the lumenal membrane leaflets and components of bacteria cell wall, thereby serving as 'eat-me' tags for autophagy. Moreover, complexes such as TRIM16-Galectin3 also mediate autophagy removal of damaged phagosomes and lysosomes during pathogen invasion (*Chauhan et al., 2016*; *Randow and Youle, 2014*). The human tumour necrosis factor receptor-associated factors (TRAF1-7) function in a number of biological processes such as innate and adaptive immunity and cell death as E3 ubiquitin ligases and scaffold proteins (*Xie, 2013*). TRAFs are composed of several domains: the N-terminal domain containing a RING motif (except TRAF1), a series of Zn-finger motifs that connect the N- and C-terminal regions and the C-terminal TRAF domain (except for TRAF7) subdivided in TRAF-N coiled-coil domain and TRAF-C domain comprised of seven to eight antiparallel $\beta$-strands (*Bradley and Pober, 2001*; *Park, 2018*; *Joazeiro and Weissman, 2000*). The RING domain and the Zn-finger motifs of TRAF6 are required for the E3 ligase activity and together with the dimeric E2-conjugating enzyme complex Ubc13-UeV1A, TRAF6 catalyses the generation of poly-Ub chains linked via the lysine K63 of Ub, in a homo- or hetero-oligomerisation-dependent manner with other TRAF members (*Middleton et al., 2017*). The

RING domain is also important for dimerization of trimeric TRAF proteins allowing the formation of signalling networks (*Das et al., 2022*). The TRAF-N domain enables TRAF proteins homo- and hetero-oligomerisation that is also crucial for their E3 ligase activity. The TRAF-C domain allows interaction with receptor and adaptor proteins (*Yin et al., 2010*). Endomembrane ruptures on one hand permit the pathogen to gain access to the cytosol, but on the other hand to be detected. Then, together with the vacuole membrane remnants, bacteria are targeted to the autophagy machinery. Defects in endomembrane repair and lysophagy correlate with weakened cellular defences against pathogens, ageing, cancer and neurodegeneration (*Papadopoulos and Meyer, 2017*). Even though the role of 'eat-me' signals in autophagy is well understood, little is known about their implication in detection and repair of pathogen-induced or sterile membrane damage. Furthermore, how pathogen and membrane damages are detected, leading to the deposition of such 'eat-me' signals, remains unknown.

Using the social amoeba *Dictyostelium discoideum – M. marinum* model system for host-pathogen interactions, we have shown previously, that upon infection, *M. marinum* does colocalize with (Ub) and the autophagy markers Atg8 and p62, and that autophagy deficiency results in drastic increase in bacteria proliferation, and that failure to repair or recycle damaged MCVs results in increased cell death (*Cardenal-Muñoz et al., 2017*; *Wang et al., 2018*; *Eriksson et al., 2020*). Here, we have unravelled a novel function of a TRAF-like family protein RING domain-containing E3 ligase. Among half a dozen candidates explored, we found that the evolutionary conserved E3-ligase TrafE is recruited to MCVs or endolysosomes in a membrane damage- or tension-dependent manner. Importantly, we provide evidence that in the absence of TrafE, the autophagy restriction of *M. marinum* as well as the autophagy-mediated and ESCRT-mediated endolysosomal membrane damage repair are severely compromised, leading to precocious cell death.

## Results

### The E3-ligase TrafE is upregulated early-on after infection and is recruited to MCVs

Using the sequence of the human TRAF6 as a reference we searched and identified bioinformatically (*Figure 1—figure supplement 1A–C*) a number of *D. discoideum* TRAF-like proteins, five of which, namely TrafA, TrafB, TrafC, TrafD, and TrafE harbour all the stereotypical TRAF protein family domains, share remarkable structural similarity with human TRAF6 (*Figure 1—figure supplement 1D, E*) and are expressed during the vegetative stage of the life cycle (*Dunn et al., 2018*; *Stajdohar et al., 2017*). In parallel, RNA-sequencing (RNA-seq) transcriptomic comparison between control and *M. marinum*-infected *D. discoideum* cells showed that out of the aforementioned candidates only *trafE* is upregulated early after infection (*Figure 1—figure supplement 2*; *Hanna et al., 2019*). TrafE is also enriched at the early MCV, as revealed by organelle proteomics (*Guého et al., 2019*).

To validate a possible role in infection of the aforementioned TRAF-like proteins, we first sought to identify candidates that display remarkable relocalization during infection. To do so, we generated stably transformed cell lines ectopically expressing TrafA, TrafB, TrafC, TrafD and TrafE candidate proteins, N- or C-terminally fused to GFP, under a constitutive promoter. We monitored the subcellular localization of these GFP fusions by live confocal microscopy over a period of 24 hr in the context of an infection with *M. marinum*, compared to a mock-infected control. As early as 1 hour post infection (hpi), GFP-TrafE showed robust recruitment to MCVs, sustained over the infection time course, predominantly in the area of *M. marinum* bacteria poles (*Figure 1A, B and C*) consistent with previously demonstrated localization of ubiquitin and the autophagy-related Atg8 (*López-Jiménez et al., 2018*). GFP-TrafA and GFP-TrafB remained solely cytosolic or nuclear, respectively, in control and infection conditions at all time points (*Figure 1—figure supplement 3A, B*). We confirmed in silico that TrafB harbours a nuclear localization signal (NLS) similarly to human TRAF4, as predicted by both NLS Mapper and NLStradamus (*Nguyen Ba et al., 2009*; *Kosugi et al., 2009*). Interestingly, despite TrafC and TrafD being 95.56% similar in terms of amino acid sequence, they displayed different behaviour upon infection. GFP-TrafC like GFP-TrafE was recruited to MCV, which was not the case for GFP-TrafD (*Figure 1—figure supplement 3C, D*).

Real-Time quantitative-PCR (RT q-PCR) carried out to monitor the transcriptional profile in control and infected cells confirmed up to fourfold upregulation of *trafE* already at 1 hpi, consistent with

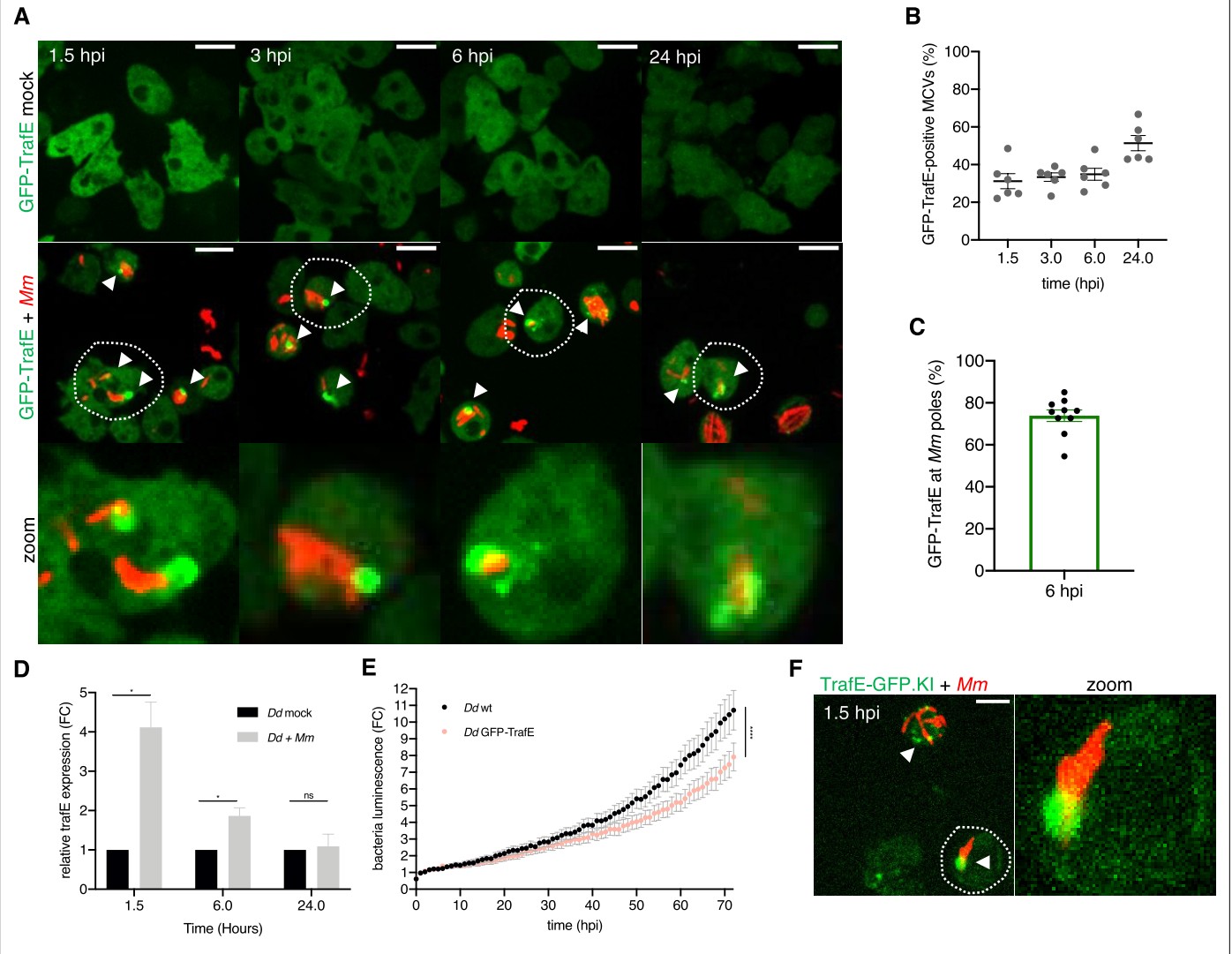

**Figure 1.** Upon infection TrafE is upregulated and recruited to MCVs. (**A**) GFP-TrafE-expressing *D. discoideum* cells (green) were mock-infected or infected and then assessed at 1.5, 3, 6, or 24 hr with mCherry-expressing *M. marinum* (red). Representative maximum projections of live time-lapse spinning disk confocal images with arrowheads pointing at GFP-TrafE recruitment to MCVs/bacteria. Scale bars correspond to 10 µm. Images are representative of at least three independent experiments. (**B**) Quantification of the percentage of intracellular MCVs/bacteria positive for GFP-TrafE during the infection time-course. Each point is representative of a 6 random multi-position field with n=2 from N=3 independent experiments. Bars represent SEM. (**C**) Quantification of GFP-TrafE in the vicinity of *M. marinum* poles from random images in N>3 independent experiments. (**D**) Normalized mRNA levels of *trafE* in mock-infected or *M. marinum*-infected *D. discoideum* cells at 1.5, 6, or 24 hr. Shown are mean and SEM of the fold change (FC) representing three independent experiments. Statistical differences were calculated with an unpaired t test (n.s.: non-significant, *: p-value ≤0.05). (**E**) The intracellular growth of *M. marinum*-lux was monitored every 1 hr inside WT and TrafE overexpressing cells, for 72 hr. Shown are mean and SEM of the fold change (FC) from three independent experiments. Statistical differences were calculated using Bonferroni multiple comparison test after ANOVA (***: p-value ≤0.001). (**F**) Recruitment of the endogenous TrafE-GFP (green) to MCV/bacteria (red) after 1.5 hpi. Scale bars correspond to 10 µm. Images are representative of at least three independent experiments.

The online version of this article includes the following source data and figure supplement(s) for figure 1:

**Source data 1.** Quantification data for *Figure 1B*.

**Source data 2.** Quantification data for *Figure 1C*.

**Source data 3.** Raw data for RT-qPCR in *Figure 1D*.

**Source data 4.** Raw data for intracellular growth assay in *Figure 1E*.

**Figure supplement 1.** Identification of *D. discoideum* TRAF-like proteins.

**Figure supplement 2.** *trafE* is upregulated early after infection.

*Figure 1 continued on next page*

*Figure 1 continued*

**Figure supplement 3.** TrafA, TrafB, TrafC, and TrafD behaviour after infection with *M. marinum*.

**Figure supplement 3—source data 1.** Quantification data for *Figure 1—figure supplement 3*.

**Figure supplement 4.** *M. marinum* growth is not affected by TrafC, TrafD overexpression and TrafE C-terminal GFP tagging.

**Figure supplement 4—source data 1.** Quantification data for *Figure 1—figure supplement 4*.

**Figure supplement 4—source data 2.** Quantification data for *Figure 1—figure supplement 4*.

the RNA-seq data (*Figure 1D* and *Figure 1—figure supplement 2*; *Hanna et al., 2019*). We also evaluated a possible overexpression phenotype of TrafC, TrafD, and TrafE during infection with bioluminescent *M. marinum* (*M. marinum*-lux)(*Sattler et al., 2007*). Whilst the intracellular growth of *M. marinum* WT remained unaffected in cells overexpressing TrafC or TrafD, this growth was significantly hampered when TrafE was overexpressed (*Figure 1—figure supplement 4A*, *Figure 1E*). These results persuaded us to explore further the role of TrafE in infection as potentially the major E3-ligase candidate.

In order to assess potential artefacts of overexpression, a TrafE-GFP chromosomal knock-in (TrafE-GFP KI) was generated and tested functionally by monitoring intracellular growth of *M. marinum*-lux. Importantly, bacteria growth in the TrafE-GFP KI strain was comparable to that in WT cells (*Figure 1—figure supplement 4B*), indicating that C-terminal GFP fusion of TrafE does not interfere with its function. In addition, the endogenous TrafE-GFP colocalized with MCVs already 1.5 hr after infection (*Figure 1F*) like the ectopically expressed GFP-TrafE (*Figure 1A*). Thus, our data points to TrafE involvement in the early MCV membrane damage detection and the MCV or *M. marinum* ubiquitination, necessary for repair or degradation. We conclude that TrafE is a conserved E3-ligase upregulated and recruited to the *M. marinum* compartment upon infection and likely involved in *D. discoideum* cell-autonomous defence.

## Loss of TrafE promotes *M. marinum* early release and replication, and is highly toxic for host cells

To explore further whether TrafE is either involved in the host defence response or is exploited by bacteria to favour their proliferation, we generated cell lines in which the *trafE* coding sequence (CDS) was ablated (*trafE*-KO), and strains overexpressing TrafE. We observed that the intracellular load of *M. marinum*-lux was increased in *trafE*-KO cells in a way comparable to the growth of the pathogen in the autophagy-impaired *atg1*-KO mutant cells (*Figure 2A*). This result, together with the decreased *M. marinum* growth in WT cells overexpressing TrafE (*Figure 1E*), indicate that the endogenous amount and activity of TrafE is limiting for bacterial restriction.

It has been demonstrated that *atg1*-KO cells suffer from a complete loss of autophagy resulting in the absence of xenophagy and thus were strongly impaired in the restriction of cytosolic bacteria (*Cardenal-Muñoz et al., 2017*; *López-Jiménez et al., 2018*). To examine whether absence of TrafE also results in a complete autophagy block or in a more specific and limited phenotype, we monitored the progression of the two mutant strains through the developmental cycle (*Otto et al., 2004*). Upon starvation, *atg1*-KO cells displayed severe developmental phenotypes and did not proceed beyond aggregation. In contrast, *trafE*-KO cells successfully completed their developmental cycle and formed viable spores (*Figure 2—figure supplement 1A*), suggesting that TrafE may not function as a general autophagy regulator but may be more specifically linked to infection- or membrane damage-related activities.

Earlier studies also indicated that in the absence of autophagy (e.g. in *atg1*-KO cells) or ESCRT-III-mediated membrane damage repair (e.g. in *tsg101*-KO cells), bacteria escape the MCV precociously to the cytosol (*Cardenal-Muñoz et al., 2017*). As a reporter for *M. marinum* escape to the cytosol, we monitored Plin, the *D. discoideum* homolog of the mammalian perilipins known to bind the hydrophobic cell wall of mycobacteria exposed to the cytosol (*Barisch et al., 2015*; *López-Jiménez et al., 2018*). The proportion of mCherry-Plin-positive bacteria indicated that, in *trafE*-KO host cells, *M. marinum* was exposed to the cytosol significantly earlier compared to WT cells, but similarly to *atg1*-KO cells (*Figure 2B*, *Figure 2—figure supplement 1B*). Importantly, in *atg1*-KO and *tsg101*-KO cells, *M. marinum* escapes precociously to the cytosol but has distinct and even opposite fates. In *atg1*-KO cells, cytosolic bacteria accumulate Ub but proliferate significantly due to a lack of

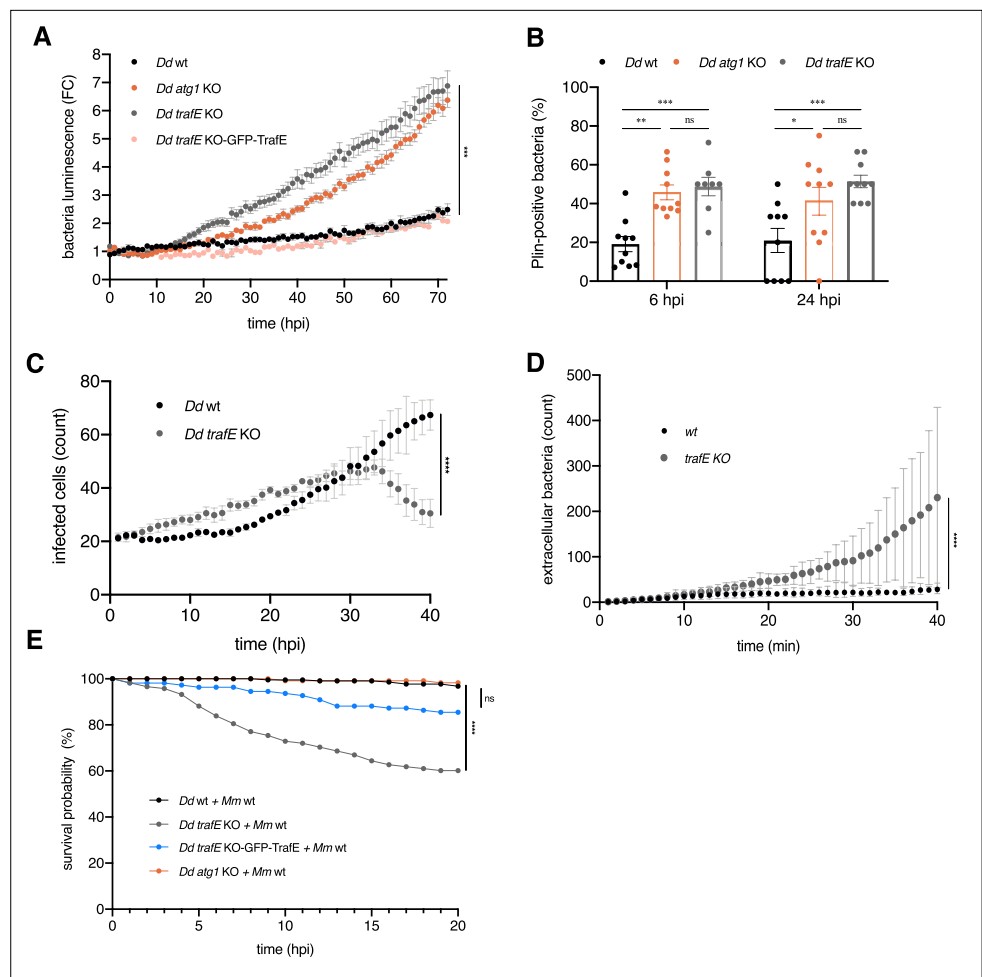

**Figure 2.** Loss of TrafE is detrimental for *D. discoideum* cells after infection with *M. marinum*. (**A**) The intracellular growth of *M. marinum*-lux was monitored every 1 hr inside WT, *atg1*-KO, *trafE*-KO or GFP-TrafE complemented *trafE*-KO cells, for 72 hr. Shown are mean and SEM of the fold change (FC) representing three independent experiments. Statistical differences were calculated using Bonferroni multiple comparison test after ANOVA ***: p-value ≤0.001. (**B**) WT, *atg1*-KO or *trafE*-KO cells expressing mCherry-Plin were infected with GFP-expressing *M. marinum*. The percentage of cells with intracellular bacteria colocalizing with Plin was assessed manually at 6 and 24 hpi. Error bars indicate the SEM from 3≤n ≤ 4 relicates from N=3 independent experiments. Statistical differences are indicated with an asterisk and were calculated with an unpaired *t* test (n.s.: non-significant, *: p-value ≤0.05, **: p-value ≤0.01, ***: p-value ≤0.001). (**C–D**) WT or *trafE*-KO cells infected with DsRed-expressing *M. marinum* imaged by high-content microscopy every 1 hr. Number of infected cells (**C**) or number of extracellular bacteria (**D**) were monitored and the average of three replicates n≥200 cells per time point was plotted. Error bars represent SEM. Cell strain-time dependent statistical differences were calculated using Bonferroni multiple comparison test after two-way ANOVA (**: p-value ≤0.01). (**E**) WT, *atg1*-KO, *trafE*-KO or GFP-TrafE-complemented *trafE*-KO cells infected with mCherry-expressing *M. marinum* in microfluidic chip single-cell experiment were monitored every 1 hr. The number of live cells was counted at each time point for 20 hr and the counts from N=3 independent experiments were plotted as Kaplan-Meier probability of survival curves. Statistical differences were calculated using Bonferroni multiple comparison correction after two-way ANOVA (n.s.: non-significant, ***: p-value ≤0.001).

The online version of this article includes the following source data and figure supplement(s) for figure 2:

**Source data 1.** Raw data for intracellular growth assay in *Figure 2A*.

**Source data 2.** Quantification data for *Figure 2B*.

**Source data 3.** Raw data for high-content segmentation in *Figure 2C*.

**Source data 4.** Raw data for high-content segmentation in *Figure 2D*.

**Source data 5.** Quantification data for single-cell experiment in *Figure 2E*.

*Figure 2 continued on next page*

*Figure 2 continued*

**Figure supplement 1.** Development of *D. discoideum* (**A**); *M. marinum* escapes early rom the MCV to the cytosol in *trafE*-KO cells (**B**).

functional xenophagy. In *tsg101*-KO cells *M. marinum* is ubiquitinated and delivered to the autophagy machinery, leading to higher restriction of bacterial growth than in WT cells (*López-Jiménez et al., 2018*). Interestingly, similarly to autophagy deficient *atg1*-KO cells, in *trafE*-KO host cells *M. marinum* escapes earlier and displays high proliferation, suggesting that both MCV damage repair and xenophagy restriction of bacteria are impaired. To monitor this, we infected WT and *trafE*-KO cells with mCherry-expressing bacteria and monitored the infection progression by high-content (HC) confocal microscopy. Surprisingly, after 30 hpi we observed a significant decrease in the number of infected *trafE*-KO cells compared to infected WT cells (*Figure 2C*). This was paralleled by a high number of extracellular *M. marinum* that had likely been released by lysis from host *trafE* mutant cells near 30 hpi (*Figure 2D*). In conclusion, we hypothesized that *M. marinum* escapes precociously to the cytosol of *trafE*-KO cells, proliferate unrestrictedly and, as a likely consequence, kills its host faster.

To validate this hypothesis, we employed a microfluidic device, the InfectChip (*Delincé et al., 2016*), in which single infected cells are trapped to allow long-term recording of host and pathogen fate (*Mottet et al., 2021*). We confirmed that *M. marinum* is highly toxic for *trafE*-KO cells, a phenotype almost fully complemented by GFP-TrafE overexpression (*Figure 2E*). Surprisingly, the survival probability of infected autophagy-deficient *atg1*-KO host cells was not significantly affected under these conditions and time-frame (*Figure 2E*), even though *M. marinum* intracellular growth is also drastically increased in these mutants compared to WT (*Cardenal-Muñoz et al., 2017*). Increased cell death likely occurs later in *atg1*-KO (*López-Jiménez et al., 2019*) compared to *trafE*-KO cells. Altogether, these results reveal that as previously observed in *atg1*-KO cells, absence of TrafE leads to *M. marinum* early escape from the MCV to the cytosol, and strongly affects its targeting by the xenophagy machinery. Importantly, the acute toxicity of *M. marinum* in *trafE*-KO cells indicates that, in contrast to Atg1, TrafE is maybe more broadly involved in cell survival mechanisms.

## TrafE action is triggered by *M. marinum*-induced MCV membrane damage

Previously, we have shown that mycobacteria escaping to the cytosol are captured and restricted by xenophagy (*López-Jiménez et al., 2018*). Taking into account that MCV membrane disruption prompts the recruitment of both the ESCRT and the autophagy machineries, we hypothesized that TrafE colocalization with *M. marinum* depends on membrane damage. Both, *M. tuberculosis* and *M. marinum* share the genomic locus Region of Difference 1 (RD1) encoding the ESX-I (type VII) secretion system required for the secretion of the membranolytic virulence factor EsxA and its chaperone EsxB (*Hagedorn et al., 2009*). As previously shown (*Cardenal-Muñoz et al., 2017*), in *D. discoideum* cells infected with *M. marinum* ΔRD1, which induces less MCV damages, ubiquitination of bacteria and their compartment is significantly decreased. In addition, nearly 80% of GFP-TrafE recruitment to MCVs was observed in the vicinity of bacteria poles (*Figure 1C*), where ESX-1 is localized and active in both *M. tuberculosis* and *M. marinum* (*Carlsson et al., 2009*). Interestingly, upon infection with *M. marinum* ΔRD1, TrafE-GFP recruitment to MCV was severely reduced (*Figure 3A and B*). Previously, we showed that both intracellular growth of WT *M. marinum* as well as autophagosome formation depend on a functional ESX-1 system. Consequently, the bacteria load of *M. marinum* ΔRD1 was significantly and similarly reduced in both WT and *atg1*-KO cells (*Cardenal-Muñoz et al., 2017*). Therefore, we measured the intracellular growth of WT and *M. marinum* ΔRD1 bacteria. Lack of the ESX-1 system drastically reduced bacterial proliferation in both WT and *trafE*-KO cells (*Figure 3C*), indicating that the growth advantage of WT *M. marinum* in *trafE*-KO cells is strictly dependent on MCV damage and escape to the cytosol. In addition, the same microfluidic device for long-term recording of single infected cells (*Figure 2E*) also revealed that the survival probability of both *trafE*-KO and *atg1*-KO host cells was not affected during infection with *M. marinum* ΔRD1 (*Figure 3D*). In conclusion, we propose that membrane damage is required for TrafE recruitment to MCVs and subsequent restriction of bacteria proliferation by xenophagy.

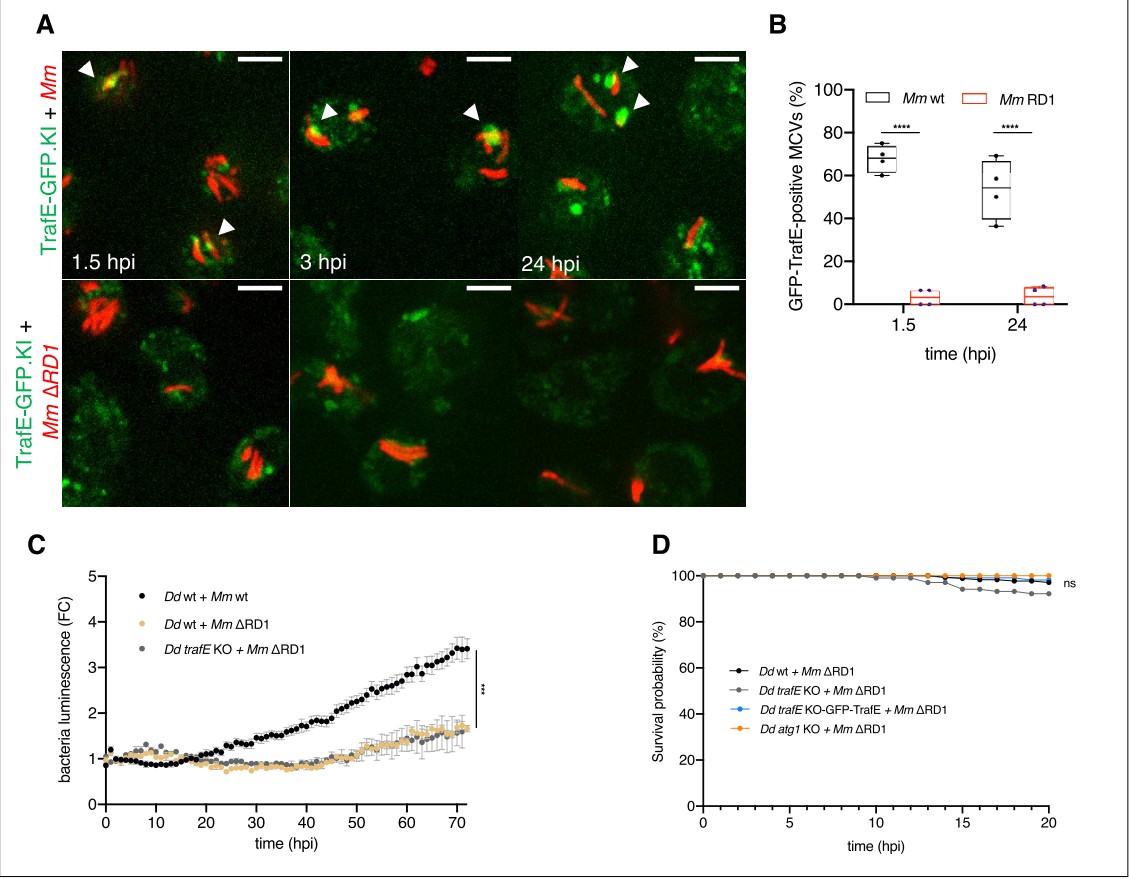

**Figure 3.** TrafE recruitment to MCVs/bacteria is membrane damage-dependent. (**A**) *D. discoideum* cells expressing endogenous TrafE-GFP (green) were infected and then assessed at 1.5, 3, or 24 hr with *M. marinum* WT (red) or *M. marinum* ΔRD1 (red). Representative maximum projections of live images with arrowheads pointing at TrafE-GFP recruitment to MCVs/bacteria. Scale bars correspond to 10 µm. Images are representative of three independent experiments. (**B**) Quantification of the percentage of intracellular MCV/bacteria positive for TrafE-GFP during the infection time-course. SEM from two to four independent experiments. Statistical differences were calculated with an unpaired t test (****: p-value ≤0.0001). (**C**) The intracellular growth of *M. marinum*-lux WT or *M. marinum* ΔRD1 was monitored every hour inside WT or *trafE*-KO cells, for 72 hr. Shown are mean and SEM of the fold change (FC) representing three independent experiments. Statistical differences were calculated using Bonferroni multiple comparison test after one-way ANOVA (***: p-value ≤0.001). (**D**) WT, *atg1*-KO, *trafE*-KO, or GFP-TrafE-complemented *trafE*-KO cells infected with *M. marinum*-lux ΔRD1 in microfluidic chip single-cell experiment were monitored every 1 hr. The number of live cells was counted at each time point for 20 hr and the counts from three independent experiments were plotted as Kaplan-Meier survival curves. Statistical difference were calculated using Bonferroni multiple comparison correction after ANOVA (n.s.: non-significant).

The online version of this article includes the following source data for figure 3:

**Source data 1.** Quantification data for *Figure 3B*.

**Source data 2.** Raw data for intracellular growth assay in *Figure 3C*.

**Source data 3.** Quantification data for single-cell experiment in *Figure 3D*.

## K63-ubiquitination levels are decreased in the absence of TrafE

A hallmark of damaged MCVs is ubiquitination known to play an essential role in pathways such as cell survival (e.g. autophagy), intracellular pathogen clearance (e.g. xenophagy) and cell death (e.g. apoptosis, necrosis)(*Cardenal-Muñoz et al., 2017*; *Papadopoulos and Meyer, 2017*; *Yoshida et al., 2017*; *López-Jiménez et al., 2018*; *Gómez-Díaz and Ikeda, 2019*; *Papadopoulos et al., 2020*). We carried out immunofluorescence assays using antibodies specific to K63-linkage polyubiquitin and determined that GFP-TrafE colocalized with K63-Ub on MCVs (*Figure 4A*). Next, we compared the total levels of polyubiquitin-positive MCVs or K63-linked polyubiquitinylated MCVs in *M. marinum*-infected WT or *trafE*-KO cells, using a global anti-polyubiquitin antibody or the specific K63-linkage polyubiquitin antibody. Interestingly, at 6 hpi the global levels of polyubiquitination between *trafE*-KO and WT cells remained comparable (*Figure 4B and C*), while the fraction of K63-linkage positive

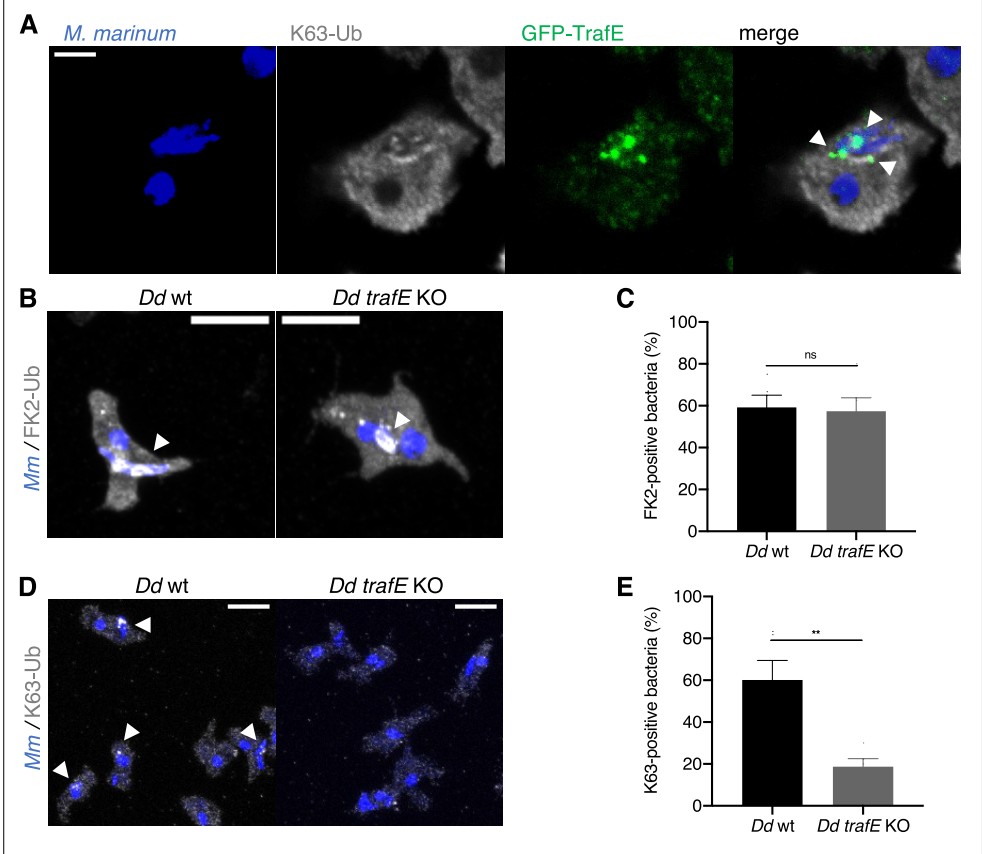

**Figure 4.** K-63-linked polyubiquitination decrease in *D. discoideum trafE*-KO cells. (**A**) Maximum projections showing colocalization (white arrowheads) of *M. marinum* (blue), K63-linked polyubiquitin chains (grey) and GFP-TrafE (green). Maximum projections showing (**B**) polyubiquitin (grey) or (**D**) K63-linked polyubiquitin (grey) colocalization (white arrowheads) with *M. marinum*. Scale bars correspond to 10 µm. Quantification of the percentage of (**C**) FK2-positive or (**E**) K63-positive intracellular MCV/bacteria at 6 hpi was carried out manually from 10 to 15 images per experiment n≥200 cells. Error bars indicate SEM. Statistical differences were calculated with an unpaired t test (n.s.: non-significant, **: p-value ≤0.01).

The online version of this article includes the following source data for figure 4:

**Source data 1.** Quantification data for immunofluorescence in *Figure 4C*.

**Source data 2.** Quantification data for immunofluorescence in *Figure 4E*.

---

MCVs was drastically reduced in *trafE*-KO cells (*Figure 4D and E*), suggesting that TrafE is involved in K63-linked polyubiquitin tagging of MCVs and bacteria.

## TrafE recruitment to bacteria is mediated by its RING domain

It has been previously reported, that TRAF E3-ligase activity depends on the RING domain that mediates the interaction between the E2-conjugating enzymes and their substrates, and TRAF-N and RING domains-dependent homo- or hetero-oligomerizations with other members of the TRAF family (*Das et al., 2021*; *Das et al., 2022*). Therefore, in order to dissect the functional importance of the RING and the TRAF-N regions during *M. marinum* infection, we generated truncated versions of TrafE that were expressed in the *trafE*-KO background (*Figure 5—figure supplement 1A, B*). Absence of the N-terminal RING domain resulted in complete loss of TrafE recruitment to MCVs throughout the infection time-course, in contrast to the truncated TRAF-N domain version or the full-length protein (*Figure 5A and B*). Despite that, functional assays using *M. marinum*-lux indicated that both truncated proteins failed to complement the *trafE*-KO (*Figure 5C*). We reason that the absence of RING domain phenocopies the truncation of the TRAF-N region because both are required for the TrafE E3 ligase activity. Additionally, the RING domain permits dimerization of TRAF trimers and formation

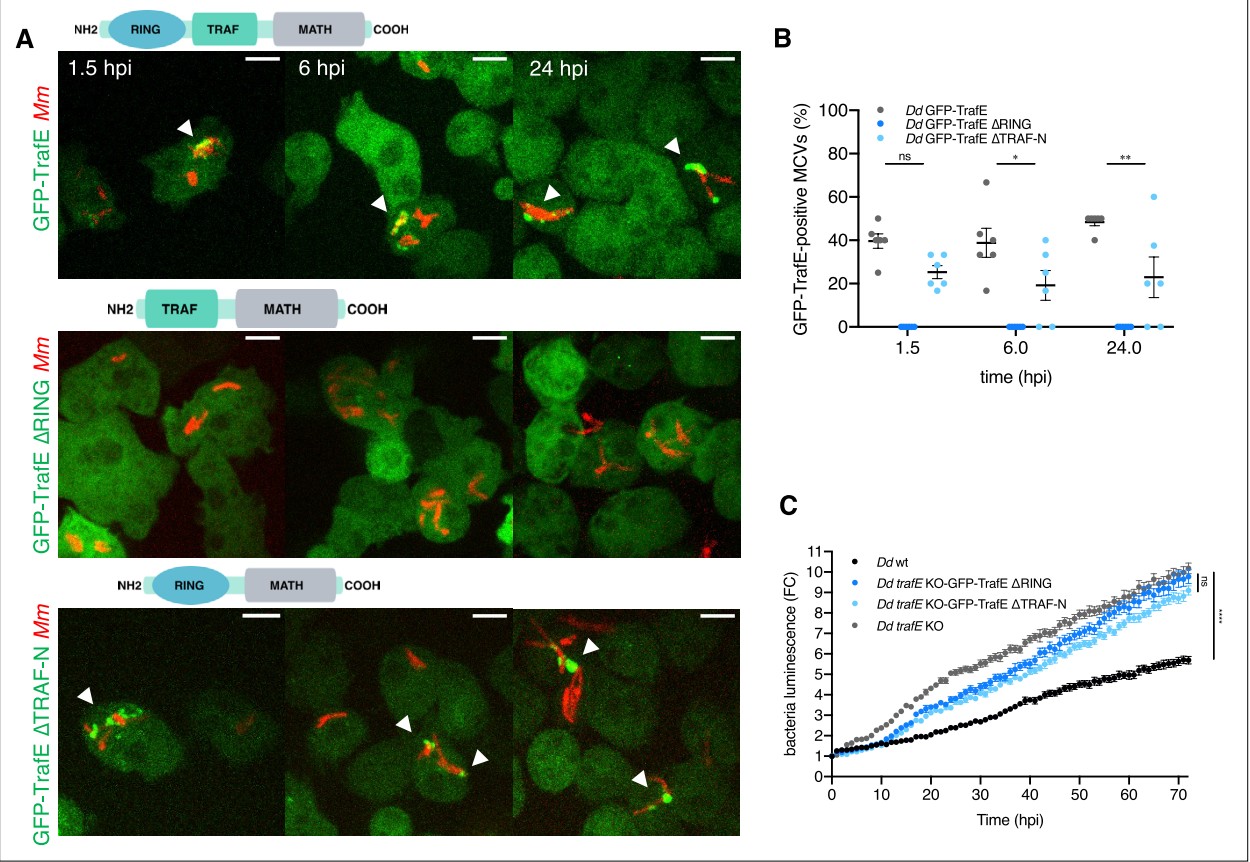

**Figure 5.** TrafE recruitment to MCV is RING domain-dependent. (**A**) *D. discoideum trafE*-KO cells expressing GFP-TrafE, GFP-TrafE with truncated RING domain or GFP-TrafE with truncated N-terminal TRAF domain (green) were infected and then assessed at 1.5, 3, 6 or 24 hr with mCherry-expressing *M. marinum* (red). Representative maximum projections of live images with arrow heads pointing at GFP- TrafE recruitment to MCVs/bacteria. Scale bars correspond to 10 μm. (**A**) Images and (**B**) quantifications are representative of three independent experiments. (**C**) The intracellular growth of *M. marinum*-lux was monitored for 72 hr, every 1 hr inside WT, *trafE*-KO or *D. discoideum* cells expressing GFP-TrafE with truncated RING domain or GFP-TrafE with truncated N-terminal TRAF domain. Shown are mean and SEM of the fold change (FC) from three independent experiments. Statistical differences were calculated using Bonferroni multiple comparison correction after ANOVA (n.s. : non-significant, *** : p value≤ 0.001).

The online version of this article includes the following source data and figure supplement(s) for figure 5:

**Source data 1.** Raw data for infection quantification in *Figure 5B*.

**Source data 2.** Raw data for intracellular growth assay in *Figure 5C*.

**Figure supplement 1.** Multiple sequence alignment (**A**) and domains and respective deletions (**B**).

---

of networks (*Das et al., 2022*), and also determines substrate recognition (*Jackson et al., 2000*) and binding to E2-conjugating enzymes, which might explain why in its absence, TrafE is not recruited to damaged MCVs.

## Sterile damage triggers TrafE recruitment on endolysosomes

Late endosomes and lysosomes are involved in the degradation of molecules endocytosed from the extracellular space or captured by the autophagy machinery in the cytosol (*Huotari and Helenius, 2011*). In *D. discoideum*, autophagy and ESCRT cooperate to promote membrane repair, however, extensively damaged endolysosomes are targeted to the autophagy machinery for removal (*López-Jiménez et al., 2018*). To address whether TrafE recruitment to bacteria is specific to infection-induced MCV damage or represents a general reaction to membrane damage, we employed the esterified dipeptide named L-leucyl-L-leucine methyl ester (LLOMe) which perforates acidic endolysosomes, making them permeable to protons and small molecules of less than 10 kDa, resulting in endolysosomal pH increase and leading to apoptosis at concentrations above 4 mM in HeLa cells (*Repnik et al., 2017*). Surprisingly, minutes after treatment with 4.5 mM LLOMe, the homogenous cytosolic

distribution of GFP-TrafE changed dramatically to dispersed puncta and rings (*Figure 6A and D*, *Figure 6—video 1*). The cytosolic GFP-TrafE ΔTRAF-N was slowly recruited on the LLOMe-damaged compartments, but the cytosolic pattern of GFP-TrafE ΔRING remained unchanged until the end of the monitoring period (*Figure 6B, C and D*). A large fraction of TrafE was clearly recruited to Alexa Fluor 647-conjugated 10 kDa dextran-loaded endolysosomes (*Figure 6E and F*, *Figure 6—video 2*). Our data suggest that the RING domain is essential for TrafE recruitment to damaged membranes, consistent with the function of TRAF6 RING domain in formation of signalling networks (*Das et al., 2022*).

## TrafE is essential for the maintenance of endolysosomal integrity

Previous results indicated that defects in the ESCRT and autophagy machineries, both known to contribute to membrane repair, lead to increased proton leakage upon LLOMe-induced sterile damage. To monitor how LLOMe affects the endolysosomal integrity and pH in the context of *trafE*-KO, we used lysosensor DND-189 which is a fluorescent pH tracer that accumulates in acidic compartments, where protonation relieves quenching. The addition of 4.5 mM LLOMe was followed by a decrease of lysosensor fluorescence intensity in both WT and *trafE*-KO cells, indicating proton leakage and pH increase within the endolysosomal compartments (*Figure 6G*). In WT cells, the lyso-sensor intensity first decreased but was restored to the initial level (*Figure 6G and H*), indicating that WT cells completed repair and recovery in the course of the 25 min of LLOMe treatment. In contrast, the lysosensor fluorescence intensity remained low in the *trafE* mutant cells (*Figure 6G and H*), indicating a more pronounced and perhaps irreversible proton leakage. The results show that cells can fully repair damage to endolysosomes and restore their pH, while absence of TrafE has a strong negative impact on the integrity and the repair of damaged compartments.

## TrafE is recruited to endolysosomes after hypertonic shock

LLOMe-induced membrane damage reduces membrane tension similarly to hyperosmotic conditions (*Mercier et al., 2020*). Indeed, water expulsion from the cytosol and from the endolysosomal compartments results in a membrane tension decrease that is sufficient to trigger the recruitment of Alix and the ESCRT-III component CHMP4B and subsequent formation of intralumenal vesicles (ILVs). To reveal possible mechanisms of TrafE recruitment to endolysosomes, we asked whether a membrane tension decrease might be sufficient to trigger recruitment of TrafE to endo-membranes. For this, we monitored *D. discoideum* cells expressing GFP-TrafE under hypertonic conditions, using as control GFP-Vps32, the *D. discoideum* homolog of the mammalian CHMP4B. Strikingly, similarly to GFP-Vps32 (*Figure 7—video 1*), cytosolic GFP-TrafE also relocated to endolysosomal compartments only seconds after the addition of sorbitol. In addition, when cells adapted and regained their initial shape and motility, TrafE regained a cytosolic distribution (*Figure 7A and B*). The pH-tracer lysosensor was used again to address whether the hyperosmotic shock induced membrane perforations. Addition of high 5 mM LLOMe concentration resulted in a decrease of lysosensor fluorescence intensity indicative of proton leakage from damaged endolysosomes. In contrast, under hypertonic conditions, the lysosensor fluorescence intensity remained stable, indicating that the membrane integrity of acidic compartment was uncompromised (*Figure 7C and D*). Our data suggest that a decrease of membrane tension alone is enough to trigger TrafE recruitment to endolysosomal membranes.

## Absence of TrafE confers acute susceptibility to sterile damage

In the absence of TrafE, the probability of survival of *M. marinum*-infected *D. discoideum* cells was drastically decreased compared to infected WT cells (*Figure 2A*). In addition, recent studies showed that disruption of lysosomal membrane leads to increase of cytosol acidity and necrosis or lysosome-dependent cell death (*Wang et al., 2018*). We hypothesized that the observed failure to repair damaged MCVs would correlate with an increased susceptibility of *trafE*-KO cells to severe sterile endolysosomal damage caused by LLOMe. To test this, the fluorescent DNA stain propidium iodide (PI), which penetrates dead or dying cells and intercalates and stains their DNA, but is excluded from live intact cells, was used as a reporter for lytic cell death. In vivo time-lapse confocal microscopy was carried out to compare viability of WT, *atg1*-KO and *trafE*-KO cells 55 min after treatment with high 5 mM LLOMe. Paralleling the results on infected cells (*Figure 2E*), cells lacking TrafE showed a highest susceptibility to LLOMe-induced sterile perforations (*Figure 7E*, *Figure 7—video 2*). These

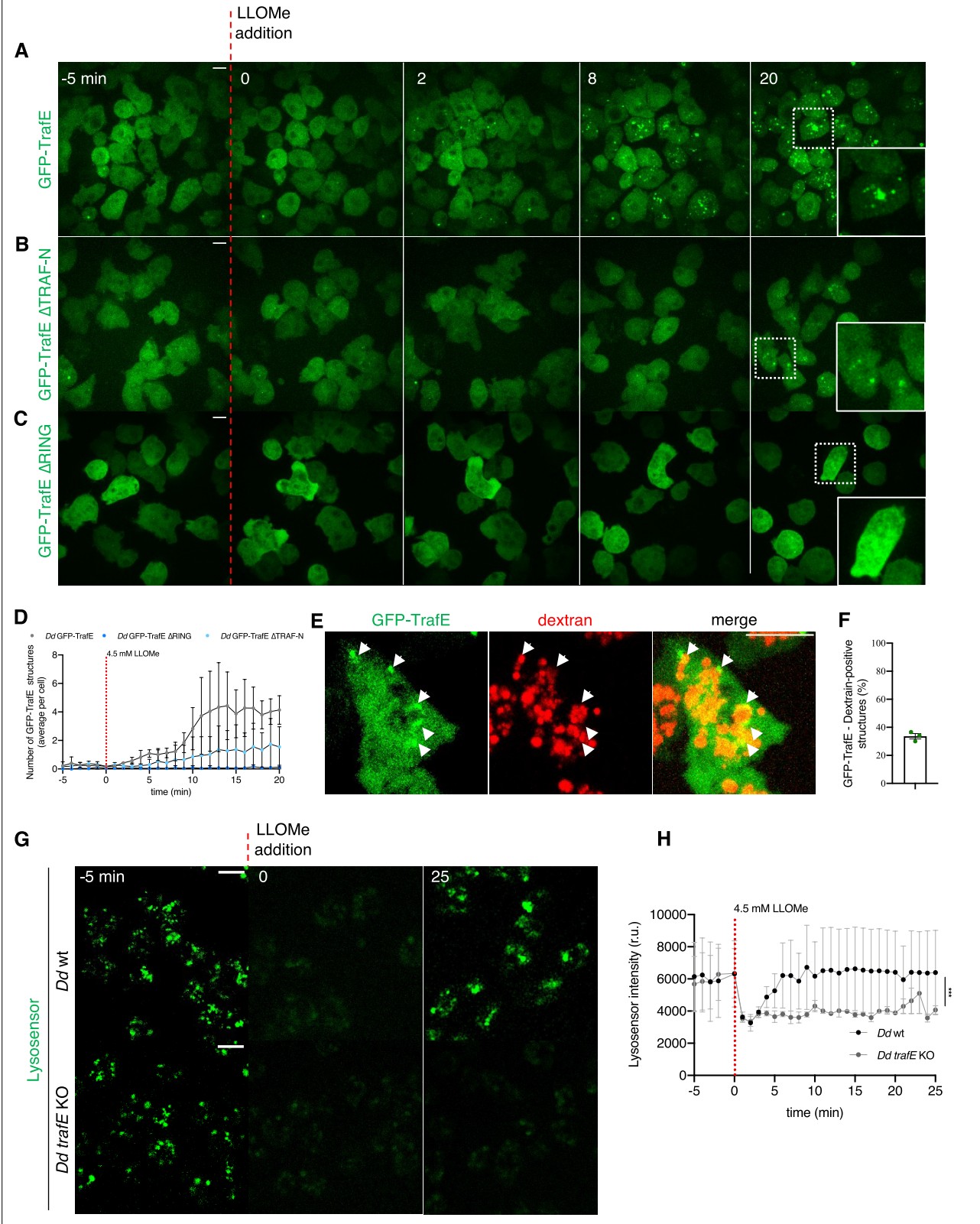

**Figure 6.** LLOMe-induced damage triggers TrafE recruitment to endolysosomes. *D. discoideum trafE*-KO cells expressing (**A**) GFP-TrafE, (**B**) GFP-TrafE with truncated N-terminal TRAF domain or (**C**) GFP-TrafE with truncated RING domain (green) were monitored by spinning disk confocal microscopy each 1 min for 25 min, 5 min before (t = –5) and 20 min after addition of 4.5 mM LLOMe at (t=0). White rectangles indicate zoom of selected areas. Images are representative of at least three independent experiments. Scale bars correspond to 10 μm. (**D**) ImageJ SpotCounter quantification of

*Figure 6 continued on next page*

*Figure 6 continued*

GFP-TrafE number of structures from N=2 independent experiments. (**E**) Colocalization of GFP-TrafE (green) with Alexa Fluor 647 10 kDa dextran (red) 10 min after LLOMe treatment. Images are representative of N>3 independent experiments. Scale bars correspond to 10 µm. (**F**) ImageJ SpotCounter count of GFP-TrafE or Alexa Fluor 647 10 kDa dextran structures followed by manual count of their colocalization from N=3 independent experiments. (**G**) Maximum projection confocal images of live *D. discoideum* WT or *trafE*-KO cells incubated with Lysosensor Green DND-189 (green), imaged every 1 min, 5 min before and 30 min after addition of 4.5 mM LLOMe. The plot is a representation of n=3 technical replicates out of N=3 biological replicates performed. Scale bars correspond to 10 µm. (**H**) Quantification of pH tracer Lysosensor Green DND-189 (green) fluorescence average intensity using ImageJ Time Series Analyzer V3. Error bars represent SEM. Statistical differences were calculated using Bonferroni multiple comparison correction after two-way ANOVA (***: p-value ≤0.001).

The online version of this article includes the following video and source data for figure 6:

**Source data 1.** Raw data for GFP-TrafE dot number quantification in *Figure 6D*.

**Source data 2.** Raw data for GFP-TrafE dextran colocalization in *Figure 6F*.

**Source data 3.** Raw data for lysosensor intensity quantification in *Figure 6H*.

**Figure 6—video 1.** TrafE relocalization to endolysosomal compartments upon LLOMe treatment.
https://elifesciences.org/articles/85727/figures#fig6video1

**Figure 6—video 2.** TrafE colocalizes with 10 kDa dextran upon LLOMe treatment.
https://elifesciences.org/articles/85727/figures#fig6video2

results suggest that the severity of the phenotypes caused by the absence of TrafE are a cumulative consequence of non-functional autophagy- and ESCRT-dependent membrane repair.

## Loss of TrafE affects the transcription of autophagy- and ESCRT-related gene sets

To investigate possible explanations for the rapid infection- or LLOMe-induced death of *trafE* null cells, as well as to identify genes and pathways underlying TrafE-mediated *M. marinum* restriction and endolysosomal homeostasis, we performed a genome-wide transcriptome analysis. RNA of *trafE*-KO mutant cells was collected, sequenced and compared to RNA of WT cells, revealing 283 genes with significantly differential expression (DE) (*Figure 8A*). Importantly, the steady-state levels of all of the ATG-related transcripts, including *atg1*, *atg9*, *atg8a* and the autophagy cargo adaptors *p62/sqstsm1* were clearly upregulated (*Figure 8B*). The respective genes are listed under the KEGG term 'autophagy-other', which is significantly enriched by gene set enrichment analysis (GSEA) (*Figure 8C*). Earlier studies have demonstrated that high, uncontrolled autophagy activity could be harmful for cells and may lead to accelerated cell death or promote replication and spread of intracellular pathogens or cancer cells (*Levine and Yuan, 2005*; *Ravikumar et al., 2010*). In addition, despite the fact that the read counts of the ESCRT-related genes do not show a clear overall pattern (*Figure 8B*), they indicate an upregulation of ALG-2-interacting-protein-X (*ALIX*) and downregulation of its putative interacting partner *Alg2A* (homolog of the human Apoptosis-linked-gene-2). Pathway enrichment analysis previously performed in our lab following RNA-seq transcriptomic comparison of non-infected or *M. marinum*-infected *D. discoideum* cells showed that most of the Biological Processes of genes upregulated during infection comprise those involved in damage response and cellular defence, particularly in the specific pathways involved in membrane repair (ESCRT, autophagy) and bacteria elimination (xenophagy) (*Hanna et al., 2019*). The steady-state transcriptomics comparison between WT and *trafE*-KO mutant cells reveals that lack of TrafE alone is sufficient to affect the expression of genes belonging to the autophagy and the ESCRT families, indicating that TrafE might function at the intersection of both ESCRT and autophagy pathways.

## Absence of TrafE does not impair autophagy induction

In *D. discoideum* cells, infection with *M. marinum*, LLOMe-induced sterile membrane damage or starvation trigger an increase of LC3/Atg8 puncta (*Cardenal-Muñoz et al., 2017*; *López-Jiménez et al., 2018*). To address whether TrafE is involved in the initiation of autophagy, a GFP-Atg8a fusion was monitored in starvation conditions, or during LLOMe-induced sterile damage. As expected in WT cells, 1.5 hr after a shift to the SorMC-Sorbitol starvation medium, the number of GFP-Atg8a puncta increased and to our surprise in *trafE*-KO cells the number of structures was significantly higher than in WT cells (*Figure 9A and B*), indicating that TrafE is apparently not required for autophagy initiation.

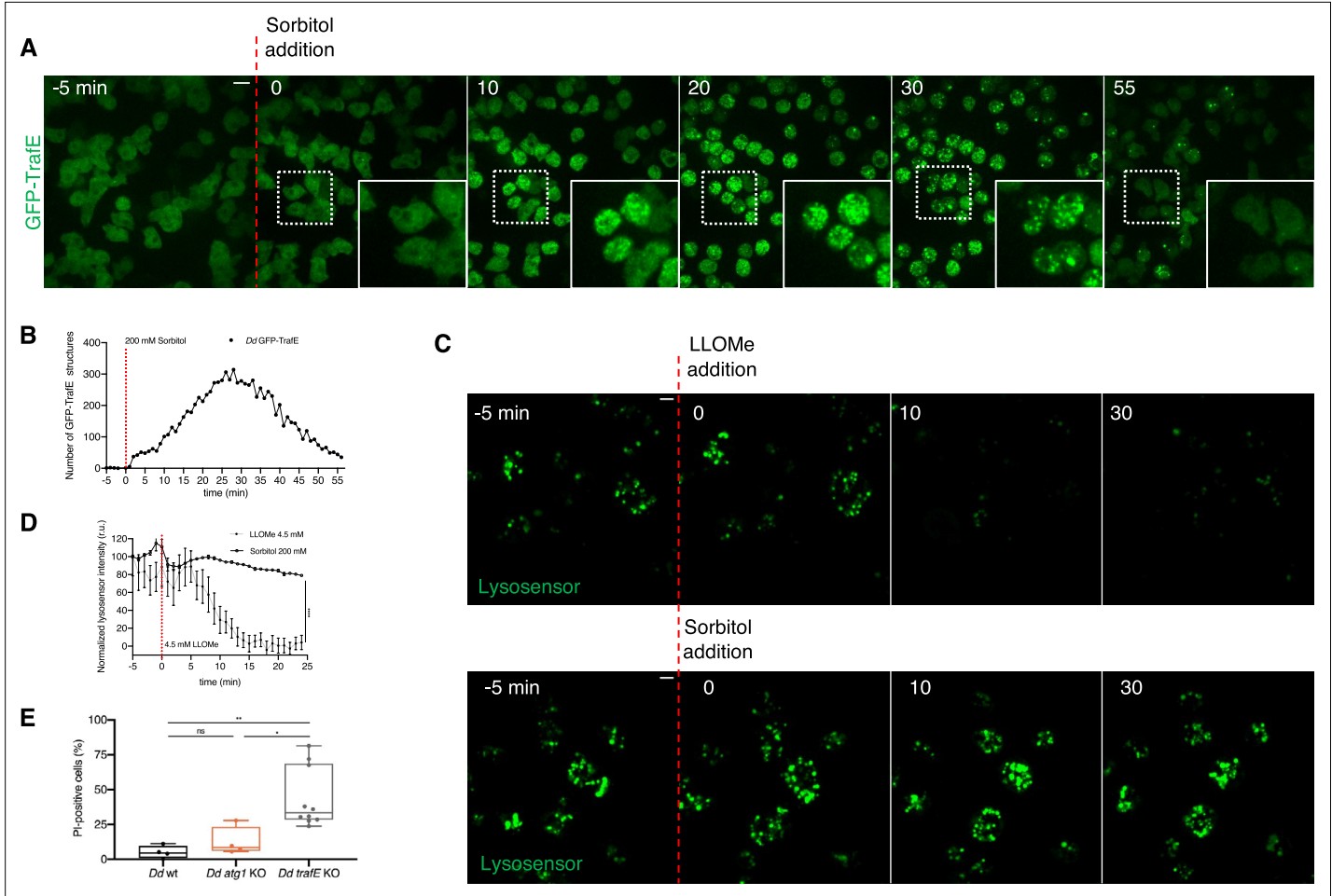

**Figure 7.** TrafE is relocalized to endolysosomes following hypertonic shock. (**A**) Maximum projection of confocal images of *D. discoideum* cells expressing GFP-TrafE (green) imaged every 1 min, 5 min before and 55 min after addition of 200 mM Sorbitol. Images are representative of at least three experiments. Scale bars correspond to 10 μm. (**B**) ImageJ SpotCounter quantification of GFP-TrafE number of structures. (**C**) Maximum projection of confocal images of live *D. discoideum* WT cells incubated with Lysosensor Green DND-189 (green) and imaged every 1 min, 5 min before and 30 min after addition of 4.5 mM LLOMe or 200 mM sorbitol. Images are representative of three independent experiments. Scale bars correspond to 10 μm. (**D**) Quantification of fluorescence average intensity of pH tracer Lysosensor Green DND-189 (green), before and after LLOMe or Lysosensor addition, using ImageJ Time Series Analyzer V3. Error bars represent SEM. Statistical difference were calculated using Bonferroni multiple comparison test after two-way ANOVA (***: p-value ≤0.001). (**E**) WT, *atg1*-KO or *trafE*-KO cells were incubated in propidium iodide (PI) and monitored by live confocal time-lapse microscopy every 1 min, 5 min before and 55 min after addition of 5 mM LLOMe. Number of PI-positive cells at 55 min were counted from 4 independent experiments. Error bars represent SEM. Statistical differences were calculated with an unpaired t test (n.s.: non-significant, *: p-value ≤0.05, **: p-value ≤0.01).

The online version of this article includes the following video and source data for figure 7:

**Source data 1.** Raw data for GFP-TrafE dot number quantification in *Figure 7B*.

**Source data 2.** Raw data for lysosensor intensity quantification in *Figure 7D*.

**Source data 3.** Quantification data for PI-positive cells in *Figure 7E*.

**Figure 7—video 1.** Vps32 is relocalized to endolysosomes following hypertonic shock.

https://elifesciences.org/articles/85727/figures#fig7video1

**Figure 7—video 2.** TrafE is required for optimal membrane damage response.

https://elifesciences.org/articles/85727/figures#fig7video2

In contrast, upon generation of sterile damage, *trafE*-KO cells exhibited a lower number of GFP-Atg8a structures compared to WT cells (*Figure 9C*). Notably, the absence of TrafE in cells treated with LLOMe lead to accumulation of large Atg8a-positive crescent-like structures (*Figure 9D*). This phenotype is reminiscent of the accumulation of Atg8/LC3 structures in yeast and mammalian cells lacking

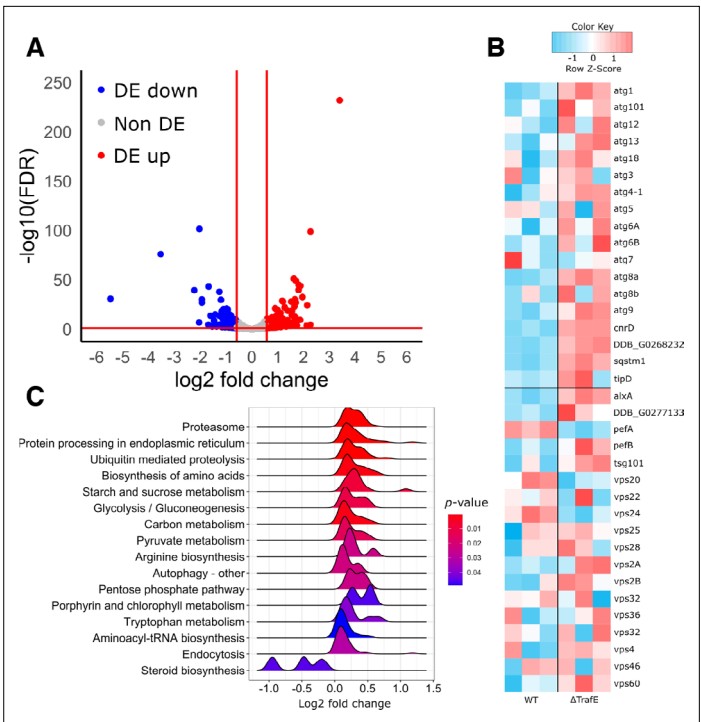

**Figure 8.** RNA-seq results. (**A**) Volcano plot of Differential Expression (DE) genes comparing *trafE*-KO to WT. After filtering for remaining ribosomal gene reads and low expressed genes, 8744 genes were left for the DE genes analysis, of which 283 were DE (absolute log2fc ≥ 0.585, FDR ≤ 0.05). From these, 144 were upregulated (red) and 139 were downregulated (blue). For illustrational purposes, *trafE* was included, which is DE and downregulated (blue dot on the far left), which is a confirmation of the *trafE* clean KO on transcriptional level. (**B**) Heatmap of normalized read counts. Filtered and normalized read counts were interrogated for a previously defined list of genes of interest. This included a set associated with autophagy (upper segment) and a set associated with the ESCRT machinery (lower segment). Biological replicates are depicted as three columns per condition (WT and *trafE*-KO). A clear pattern of upregulation is visible in autophagy-related genes, whereas no clear overall pattern is visible for ESCRT-related genes. However, key players in the ESCRT machinery such as *alxA* show a striking pattern and most notably, $Ca^{2+}$ dependent regulators *pefA* and *pefB* are regulated inversely. None of the genes depicted are DE by the statistical criteria defined previously. (**C**) Gene set enrichment analysis (GSEA) of log2 fold change of filtered genes. For GSEA, genes were ranked by log2 fold change, resulting groups were restricted by minimum and maximum group size of respectively 2 and 200 and considered significant at a *p value* ≤ 0.05 and a *q*-value ≤ 0.1. Depicted are the log2 fold change density distributions for enriched core genes in each respective term. The significant signature included several metabolic terms including Proteasome, Endocytosis and most notably Autophagy as being upregulated.

The online version of this article includes the following source data for figure 8:

**Source data 1.** RNA-seq mapped reads, DEG and GSEA OTs for *Figure 8A, B and C*.

or expressing dominant-negative forms of ESCRT subunits required for autophagosome biogenesis, closure and maturation (*Takahashi et al., 2018*; *Zhou et al., 2019*). Our data suggest that the *D. discoideum* E3-ligase TrafE acts downstream of Atg1 in the autophagy pathway. The accumulation of Atg8a puncta in starved *trafE*-KO cells is likely a consequence of a partial impairment but not a complete block of autophagy flux, because *trafE*-KO cells complete their developmental cycle unlike the fully autophagy deficient *atg1*-KO cells (*Figure 2—figure supplement 1A*).

## TrafE regulates ESCRT subunits dynamics on damaged membranes

In mammalian and *D. discoideum* cells, LLOMe-induced sterile membrane damage triggers vigorous recruitment of the ESCRT subunits involved in membrane scission, ALIX and CHMP4B/Vps32, to damaged endolysosomal compartments (*López-Jiménez et al., 2018*; *Mercier et al., 2020*). To understand better the connection between TrafE and the ESCRT pathway, we monitored the behaviour of ALIX and Vps32 GFP-fusions during LLOMe-induced sterile damage in the background of WT and

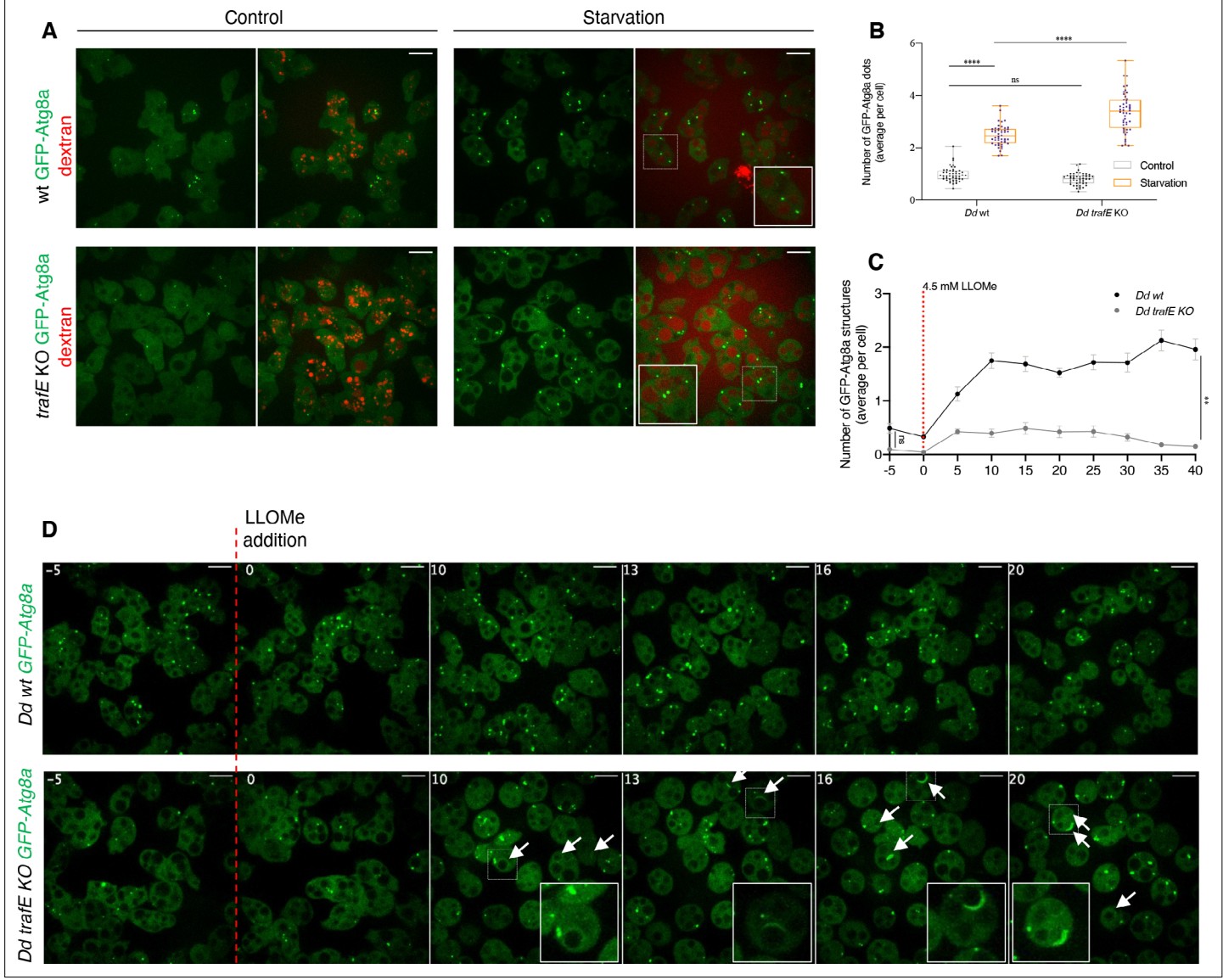

**Figure 9.** TrafE absence promotes Atg8a accumulation or formation of aberrant structures in starved or LLOMe-treated cells. (**A**) Live confocal microscopy images of *D. discoideum* WT or *trafE*-KO cells expressing *act5*::GFP-Atg8a (green) incubated for 12 hr with Alexa Fluor 647 10 kDa dextran (red) and imaged 90 min after a shift to SorMC-Sorbitol starvation medium. (**B**) Live high-content imaging quantification of *D. discoideum* WT or *trafE*-KO cells expressing *act5*::GFP-Atg8a 90 min after a shift to SorMC-Sorbitol starvation medium. GFP-Atg8a dot average number per cell was collected from 27 images with 20≤n ≤ 40 cells per image. (**C**) Quantification of live time-lapse high-content confocal images of *D. discoideum* WT or *trafE*-KO cells expressing *act5*::GFP-Atg8a, 5 min before (t = –5) and after addition of 4.5 mM LLOMe at t=0 for a total of 40 min. Images were taken every 5 min and GFP-Atg8a dot count per image was collected from 3 wells with 2 image fields per well, each image contains 20≤n ≤ 40 cells. The plot is a representation of n=3 technical replicates out of N=3 biological replicates performed. (**D**) Live time-lapse spinning disk confocal images of *D. discoideum* WT or *trafE*-KO cells expressing *act5*::GFP-Atg8a, 5 min before (t = –5) and after addition of 4.5 mM LLOMe at t=0. White arrowheads point at aberrant structures, zoomed within white rectangles. All statistical differences were calculated using Bonferroni multiple comparison correction after two-way ANOVA (n.s.: non-significant and ****: p-value ≤0.0001).

The online version of this article includes the following source data for figure 9:

**Source data 1.** Raw data for high-content segmentation in *Figure 9B*.

**Source data 2.** Raw data for high-content segmentation in *Figure 9C*.

*trafE*-KO cells. Interestingly, minutes after addition of LLOMe, as observed in mammalian cells, cytosolic ALIX-GFP was vigorously redistributed to damaged compartments (*Figure 10A*). As expected (*López-Jiménez et al., 2018*), addition of LLOMe triggered recruitment of GFP-Vps32 to damaged endolysosomes (*Figure 10B*). Most surprisingly, ALIX-GFP and GFP-Vps32 redistribution from the

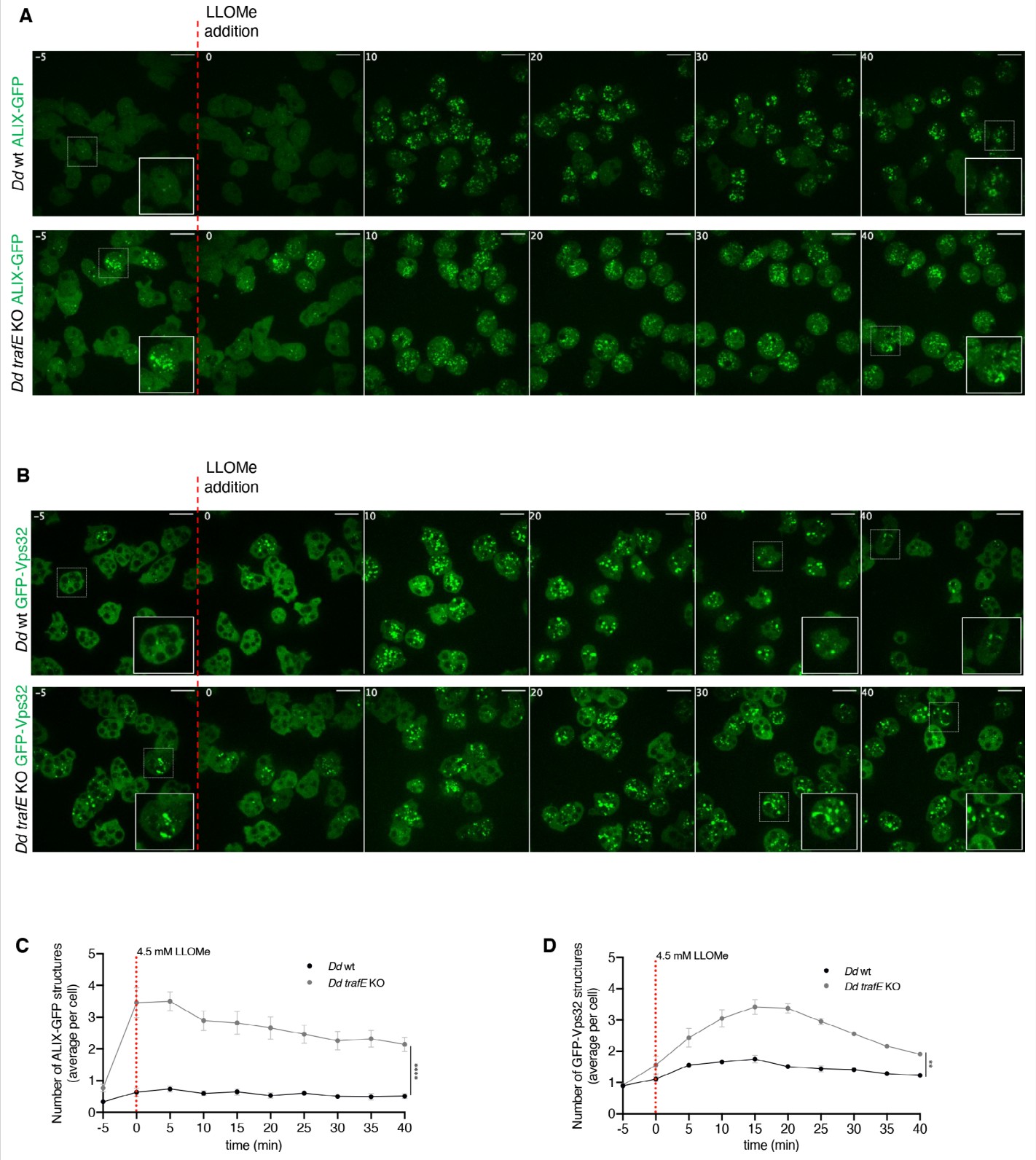

**Figure 10.** TrafE is necessary for proper ESCRT turnover during membrane repair. (**A and B**) Live time-lapse spinning disk confocal images with dedicated x100 objective of *D. discoideum* WT or *trafE*-KO cells expressing *act5*::ALIX-GFP or *act5*::GFP-Vps32, 5 min before (t = –5) and after addition of 4.5 mM LLOMe at t=0. (**C and D**) Quantification of live time-lapse high-content confocal images with dedicated x40 objective of *D. discoideum* WT or *trafE*-KO cells expressing *act5*::ALIX-GFP or *act5*::GFP-Vps32, 5 min before (t = –5) and after addition of 4.5 mM LLOMe at t=0, every 5 min for a total of

*Figure 10 continued on next page*

*Figure 10 continued*

40 min. Number of GFP-positive structures was measured from three wells with two image fields per well, each image contains 20≤n ≤ 80 cells. The plot is a representation of n=3 technical replicates out of N=3 biological replicates performed. All statistical differences were calculated using Bonferroni multiple comparison correction after two-way ANOVA (**: p-value ≤0.01).

The online version of this article includes the following source data and figure supplement(s) for figure 10:

**Source data 1.** Raw data for high-content segmentation in *Figure 10C*.

**Source data 2.** Raw data for high-content segmentation in *Figure 10D*.

**Figure supplement 1.** ESCRT machinery components redistribution upon starvation.

**Figure supplement 1—source data 1.** Quantification data for *Figure 10—figure supplement 1*.

**Figure supplement 1—source data 2.** Quantification data for *Figure 10—figure supplement 1*.

**Figure supplement 1—source data 3.** Quantification data for *Figure 10—figure supplement 1*.

**Figure supplement 1—source data 4.** Quantification data for *Figure 10—figure supplement 1*.

cytosol to damaged, dextran-positive, compartments was significantly more pronounced in *trafE*-KO compared to WT cells (*Figure 10C and D*). Note that this 'overshoot' in the recruitment of ESCRT components is counterintuitive in the light of an expected repair defect. In order to further dissect the functionality of the recruited subunits, the behaviour of these ESCRT reporters was monitored in starvation conditions. In WT and *trafE*-KO cells starved for 1.5 hr, both ALIX-GFP and GFP-Vps32 redistributed to a high number of dextran-negative foci and structures (*Figure 10—figure supplement 1A, B*), suggesting that the ESCRT machinery components are recruited to autophagosomes. This hypothesis was confirmed by the lack of a corresponding increase of ALIX-GFP or GFP-Vps32 structures in starved autophagy initiation-deficient *atg1*-KO cells (*Figure 10—figure supplement 1C, D*). Remarkably, in starved *trafE*-KO cells, the number of both ALIX-GFP or GFP-Vps32 foci and structures was significantly higher relative to WT cells (*Figure 10—figure supplement 1E, F*). The excessive accumulation of ESCRT components on damaged endolysosomes or autophagy-related structures observed in *trafE*-KO cells might reflect an incomplete engagement of both ESCRT and autophagy machineries, thereby placing TrafE at their intersection.

## Vps4 recruitment to membrane damage sites is TrafE-dependent

Previous studies have demonstrated that the Vps4 ATPase is essential for the assembly, the disassembly and the dynamic turnover of ESCRT-III subunits during MVB formation and membrane remodelling, and that the absence of Vps4 leads to accumulation of ESCRT components on endosomes (*Shestakova et al., 2010*; *Mierzwa et al., 2017*), formation of Atg8 clusters and defects in autophagosome closure (*Takahashi et al., 2018*). To investigate why absence of TrafE causes an accumulation of Atg8a, ALIX and Vps32 structures during starvation or sterile damage, we compared the behaviour of GFP-Vps4 between WT or *trafE*-KO cell lines. As expected (*López-Jiménez et al., 2018*), after addition of 4.5 mM LLOMe in WT cells, GFP-Vps4 was robustly redistributed from the cytosolic pool to damaged endolysosomes. In striking contrast, in *trafE*-KO cells the recruitment of GFP-Vps4 to endolysosomal damage sites was abolished (*Figure 11A and B*). This phenotype was reversed by re-expression of mRFP-TrafE in GFP-Vps4 expressing *trafE*-KO cells (*Figure 11—figure supplement 1*), and mRFP-TrafE colocalized with Vps4 on damaged endolysosomes (*Figure 11D*). We conclude that absence of the E3-ligase TrafE somehow impairs Vps4 recruitment to damage sites where ALIX-GFP and GFP-Vps32 accumulation results in non-functional ESCRT-mediated repair.

## In *trafE*-KO cells, the absence of Vps4 correlates with low K63-polyubiquitination despite the aberrant presence of ALIX-GFP

Previous studies have linked ALIX binding to K63-Ub chains with the establishment of a positive feedback loop resulting in the stimulation of Vps4 activity. It was demonstrated that failure of Vps4 to interact with Ub-bound ALIX results in major defects in cargo sorting (*Pashkova et al., 2013*; *Tseng et al., 2021*; *Pashkova et al., 2021*). Comparison of *D. discoideum* ALIX with a group of structurally defined but evolutionary distant coiled-coil Ubiquitin Binding Domain (UBD)-containing proteins revealed the presence of highly conserved residues, indicating a likely UBD (*Figure 12—figure supplement 1*). Therefore, to address mechanistically the functional significance of TrafE-deposited

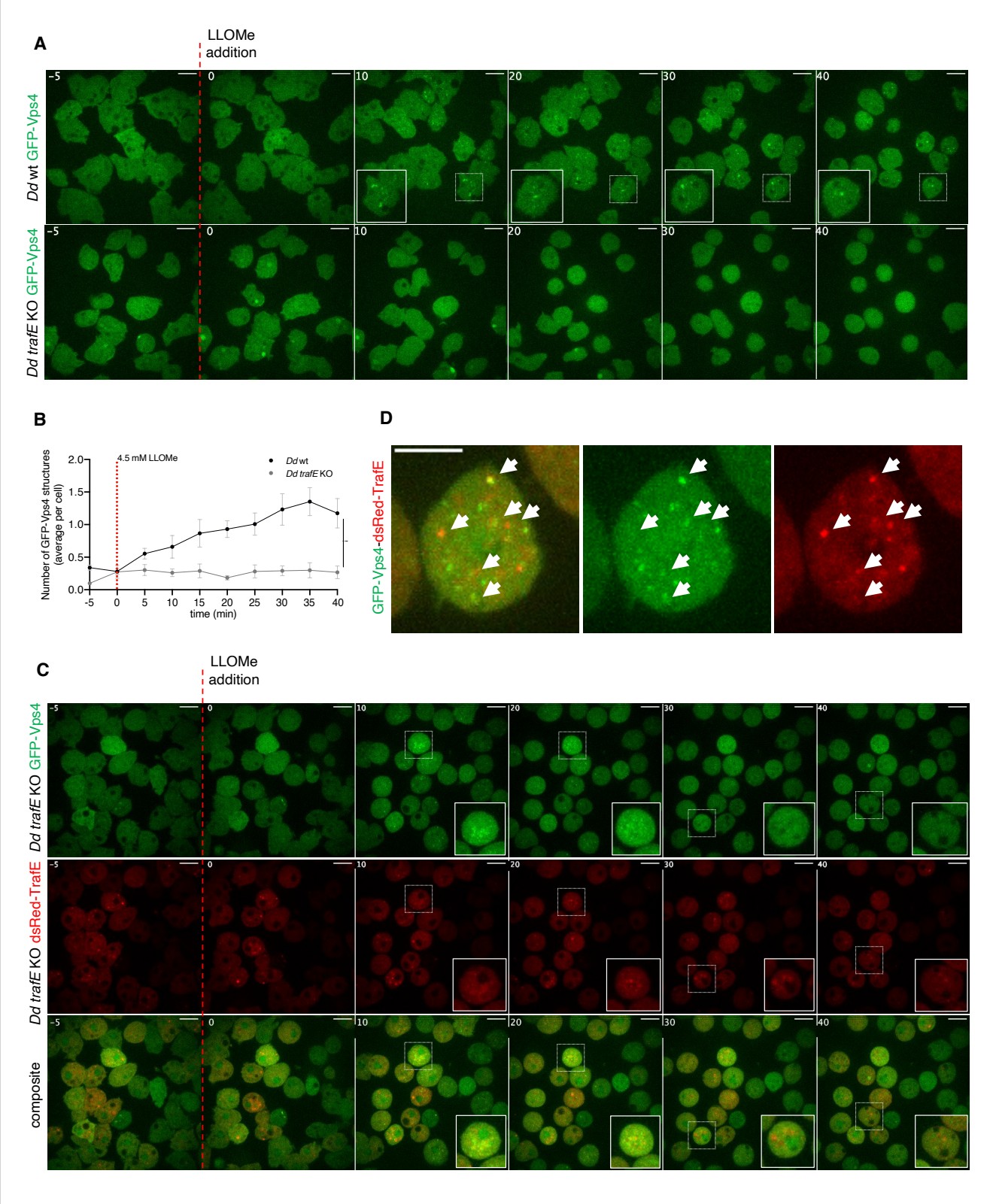

**Figure 11.** TrafE regulates Vps4 recruitment to endolysosomal membrane damage sites. (**A and B**) Live time-lapse spinning disk confocal images of *D. discoideum* WT or *trafE*-KO cells expressing *act5*::GFP-Vps4, 5 min before (t = –5) and after addition of 4.5 mM LLOMe at t=0. (**B**) Quantification of live time-lapse high-content confocal images of *D. discoideum* WT or *trafE*-KO cells expressing *act5*::GFP-Vps4, 5 min before (t = –5) and after addition of 4.5 mM LLOMe at t=0, every 5 min for a total of 40 min. Number of GFP-positive structures was measured from three wells with two image

*Figure 11 continued on next page*

*Figure 11 continued*

fields per well, each image contains 20≤n ≤ 80 cells. GFP-Vps4 puncta tend to be small and difficult to detect, therefore the represented numbers are underestimation. The plot is a representation of n=3 technical replicates out of N=3 biological replicates performed. All statistical differences were calculated using Bonferroni multiple comparison correction after two-way ANOVA (**: p-value ≤0.01). (C) Live time-lapse spinning disk confocal images of *D. discoideum* complemented *trafE*-KO cells expressing *act5*::GFP-Vps4 (green) and dsRed-TrafE (red), 5 min before (t = –5) and after addition of 4.5 mM LLOMe at t=0. (D) Live time-lapse spinning disk confocal images showing (white arrowheads) colocalization of GFP-Vps4 (green) and dsRed-TrafE (red) in LLOMe-treated *trafE*-KO cells.

The online version of this article includes the following source data and figure supplement(s) for figure 11:

**Source data 1.** Raw data for high-content segmentation in *Figure 11B*.

**Figure supplement 1.** ImageJ SpotCounter quantification of live time-lapse spinning disk confocal images of *D. discoideum* complemented *trafE*-KO cells expressing *act5*::GFP-Vps4 and dsRed-TrafE from N=3 independent biological replicates.

**Figure supplement 1—source data 1.** Quantification data for *Figure 11—figure supplement 1*.

K63-linked polyubiquitin chains during sterile damage, we carried out immunofluorescence assays of mock-treated cells or 10 min after addition of LLOMe in WT, *trafE*-KO or *atg1*-KO cells expressing ALIX-GFP together with antibodies specific to K63-polyubiquitin. As expected, compared to mock-treated, LLOMe-treated WT and *atg1*-KO cells revealed a robust redistribution of cytosolic ALIX-GFP to damaged compartments where it colocalized with K63-linked polyubiquitin (*Figure 12A, C, D and E*). Remarkably, TrafE mutants displayed the usual high number of ALIX-GFP structures in both the mock- and the LLOMe-treated conditions, however the K63-linked ubiquitination of damaged compartments was virtually abolished (*Figure 12B, D and E*).

We conclude that the *D. discoideum* E3-ligase TrafE is responsible for K63-linked polyubiquitination of MCV or endolysosomal membrane protein(s) upon damage triggered by perforation-induced changes in membrane. According to our working model in WT cells (*Figure 13A*), ALIX, Vps32 and TrafE are recruited to small size damage sites. TrafE-dependent ubiquitination of unknown endolysosomal membrane protein(s) with K63-polyubiquitin promotes K63-Ub-ALIX interaction which enhances ALIX ability to stimulate Vps4 activity and triggers its functional recruitment to damage sites resulting in proper ESCRT-III turnover. In case of substantial damage, it is possible that TrafE deposited K63-Ub serves as a signal for the recruitment of the autophagy adaptor p62 and the autophagy machinery leading to formation of a phagophore-like structure which seals by membrane fusion the damage site. In *atg1*-KO cells (*Figure 13B*), the ESCRT-dependent damage repair is functional, however the cells are autophagy deficient and therefore there is no formation of phagophore and no autophagy-dependent fusion-type membrane repair. Finally, there is no xenophagy.

In *trafE*-KO cells (*Figure 13C*), ALIX and Vps32 are recruited to damage sites; however, in the absence of ALIX interaction with TrafE-deposited K63-Ub, ALIX-dependent stimulation of Vps4 activity is insufficient to allow proper ESCRT-III turnover and functional ESCRT-dependent repair. In parallel, the absence of K63-Ub also impairs the autophagy-dependent fusion-type repair which altogether explains the severity of the *trafE*-KO phenotypes compared to *atg1*-KO mutant cells. Based on recent studies revealing the function of the ESCRT machinery in phagophore closure (*Takahashi et al., 2018*; *Takahashi et al., 2019*; *Zhen et al., 2020*), we speculate that TrafE may also be involved in this process (*Figure 13D*).

## Discussion

In recent years, studies in mammals have revealed some of the components and the effector pathways that detect, repair and / or remove intracellular compartments disrupted by invading pathogens or by mechanical damage. Cells have developed molecular sensors that determine the size and the extent of membrane damage. Recent evidence indicate that small, pore-size damage is repaired by the ESCRT machinery, however extensive, and prolonged damage triggers the autophagy machinery. The latter appears to play both a repair as well as a remove role upon catastrophic damage. Despite intense research, we still need to extend our insight in the molecular factors involved in membrane damage detection and how it is translated into a vigorous and specific ubiquitination response. *D. discoideum* is a powerful model system to address the aforementioned questions thanks to its experimental versatility and the high evolutionary conservation of the ESCRT and the autophagy pathways with their mammalian counterparts.

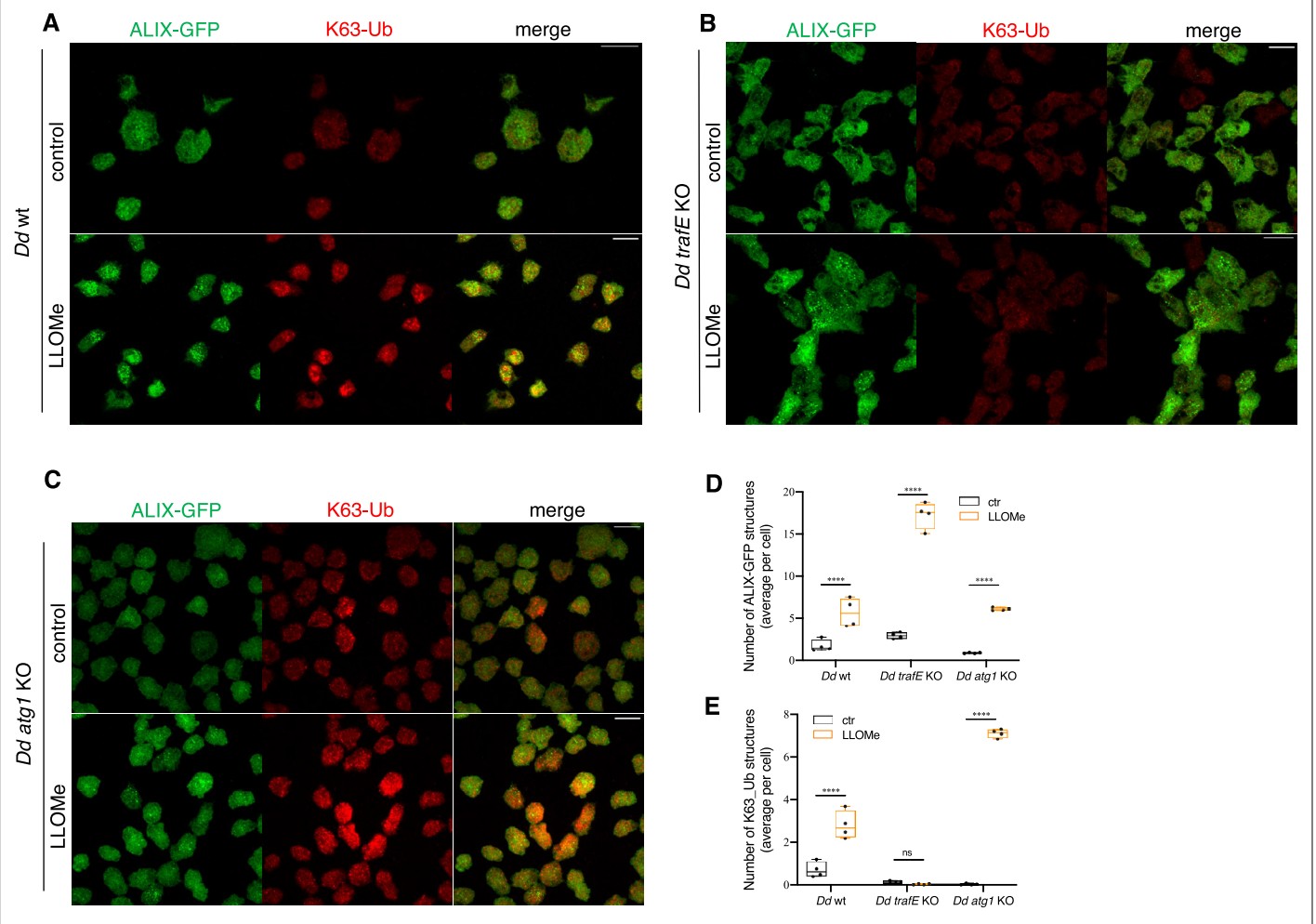

**Figure 12.** In *trafE*-KO cells, the absence of Vps4 correlates with low K63-polyubiquitination despite the accumulation of ALIX-GFP and GFP-Vps32. (**A, B and C**) Immunofluorescence images of mock-treated (mock) or 10 min after addition of LLOMe in WT, *trafE*-KO or *atg1*-KO cells expressing *act5*::ALIX-GFP (green) stained with antibodies specific to K63-polyubiquitin (red). (**D and E**) Average number of ALIX- or K63-Ub-positive structures per cells compared between mock-treated (mock) or cells treated with LLOMe for 10 min. Quantification of ALIX-GFP or K63-Ub structures from N=4 independent experiments comprised of 5≥n ≥ 45 cells per image was performed using ImageJ ValelabUtils SpotCounter. Statistical differences were calculated using Bonferroni multiple comparison correction after two-way ANOVA (n.s.: non-significant and ****: p-value ≤0.0001).

The online version of this article includes the following source data and figure supplement(s) for figure 12:

**Source data 1.** Quantification data for number of ALIX-GFP- and K63-Ub-positive structures in *Figure 12D and E*.

**Figure supplement 1.** ALIX likely comprises a UBD.

It was previously shown that in *D. discoideum*, *M. marinum* infection or LLOMe-induced sterile damage leads to an increase of Ub- and Atg8-positive structures and to recruitment of the ESCRT subunits Tsg101, Vps32 and Vps4, reportedly ahead of the recruitment of autophagy markers p62 and Atg8 at the MCV and endolysosomes (*Cardenal-Muñoz et al., 2017*; *López-Jiménez et al., 2018*). In this study, we demonstrate that early after infection, the *D. discoideum* E3 ubiquitin ligase TrafE is recruited to MCVs in a membrane damage-dependent manner (*Figures 1A, B, and 3A, B*). This recruitment was significantly attenuated upon infection with *M. marinum* ΔRD1 (*Figure 3A and B*) that produces PDIMs but lacks a functional ESX-1 secretion system. We also demonstrate that the intracellular growth of *M. marinum*-lux is significantly increased in *atg1*-KO and *trafE*-KO cells compared to WT cells (*Figure 2A*), suggesting that xenophagy is the main *M. marinum* cell-autonomous restriction pathway. In addition, cells lacking the E3 ligase TrafE show exceptionally high and early susceptibility to infection with WT *M. marinum* which is not the case in the autophagy-deficient *atg1*-KO cells (*Figure 2E*). Upon starvation, *atg1*-KO cells that are fully defective in autophagy, display severe

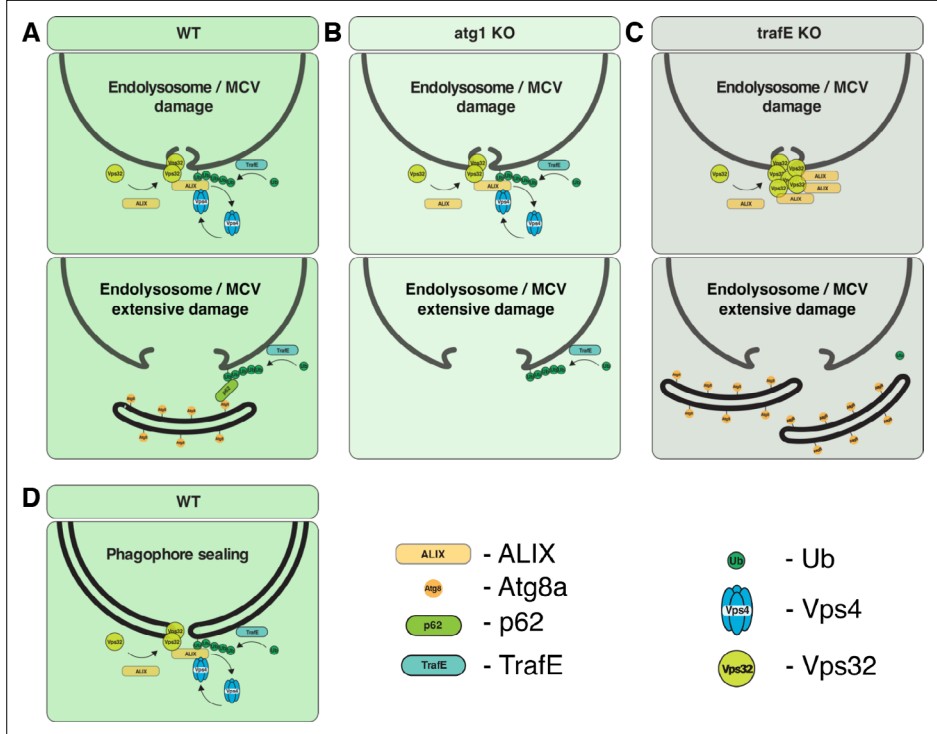

**Figure 13.** Working model. (**A**) In WT cells, ALIX, Vps32, and TrafE are recruited to small size damage sites. TrafE-dependent ubiquitination of unknown endolysosomal membrane protein(s) with K63-polyubiquitin promotes K63-Ub-ALIX interaction which enhances ALIX ability to stimulate Vps4 activity, a step necessary for its functional recruitment to damage sites resulting in proper ESCRT-III turnover. In case of substantial damage, it is possible that TrafE-deposited K63-Ub serves as a signal for the recruitment of the autophagy adaptor p62 and the autophagy machinery, leading to formation of a phagophore-like structure which seals the damage site by membrane fusion. (**B**) In *atg1*-KO cells, the ESCRT-dependent damage repair is functional; however, the cells are autophagy deficient and therefore there is no formation of phagophore, hence no autophagy-dependent fusion-type membrane repair. (**C**) In *trafE*-KO cells, ALIX and Vps32 are recruited to damage sites where the lack of ALIX interaction with TrafE-deposited K63-Ub hinders Vps4 activity stimulation and as a consequence, ESCRT-III turnover and functional ESCRT-dependent repair are impaired. In parallel, the absence of K63-Ub also impairs the autophagy-dependent fusion-type repair. (**D**) Based on recent studies revealing the function of the ESCRT machinery in phagophore closure, we speculate that TrafE may also be involved in this process.

developmental phenotypes and do not proceed beyond aggregation (*Otto et al., 2004*). In contrast, *trafE*-KO cells successfully complete their developmental cycle and form spores (*Figure 2—figure supplement 1A*), emphasizing the functional specificity of TrafE to endolysosomal membrane damage. Altogether, our data indicate that *trafE*-KO phenocopies the autophagy-deficient *atg1* mutants in terms of unrestricted *M. marinum* intracellular growth, however the higher cytotoxicity exerted by bacteria in *trafE*-KO cells compared to WT and *atg1*-KO cells (*Figures 2E and 3D*) leads to the speculation that TrafE might be upstream of both ESCRT- and autophagy-mediated membrane repair pathways. Following that hypothesis, the particularly rapid death of infected *trafE*-KO cells likely results from inefficient MCV repair, and leads to early bacteria escape and release by host cell lysis (*Figure 2B and D*, *Figure 2—figure supplement 1B*).

Ubiquitin is required for various aspects of autophagy, such as removal of protein aggregates (aggrephagy), recycling of mitochondria (mitophagy) and bacteria (xenophagy), or endolysosomal homeostasis (lysophagy). Xenophagy depends on decoration of intracellular pathogens with K63-linked Ub chains that bring bacteria to the phagophore membrane via autophagy cargo receptors such as p62 and NDP52. We demonstrated that although the levels of polyubiquitin-positive MCVs remain unaffected in *M. marinum*-infected *trafE*-KO cells, the levels of K63-linked polyUb-positive MCVs are significantly decreased but not abolished compared to wt *D. discoideum* cells (*Figure 4A, B, C, D and* ), possibly due to the presence of other E3-ligases, such as for instance TrafC, which is also

recruited to MCVs during infection and might have partially redundant functions (*Figure 1—figure supplement 3C*). It has been shown that human TRAF6 forms heterodimers with TRAF2, TRAF3, and TRAF5 through their RING domains and that the resulting heterodimers are more stable than the TRAF6 homodimers (*Das et al., 2021*). Therefore, the severity of the *trafE*-KO phenotypes during infection may be due to the instability or non-functionality of a potential heterodimerization redundant partner(s), such as other *D. discoideum* TRAFs. Furthermore, we found that the recruitment of TrafE to MCVs is abolished in the absence of its RING domain and not when the TRAF domain is deleted (*Figure 5A and B*), likely because RING domains-mediated dimerization confers the specificity to the substrate via TrafE interaction with the E2-conjugating enzyme (*Middleton et al., 2017*; *Kiss et al., 2021*). Nevertheless, both domains are equally important for the restriction of *M. marinum* intracellular growth (*Figure 5C*).

Similar to xenophagy, lysophagy also depends on Ub and it has been shown that ubiquitination of Lysosome-Associated Membrane Protein 2 (LAMP2) in damaged lysosomes leads to their autophagic degradation (*Papadopoulos et al., 2017*). In addition, recent studies in mammalian cells have demonstrated that disrupted lysosomes leak protons and reactive oxygen species that lead to pyroptotic cell death (*Papadopoulos et al., 2020*). Repair of damaged lysosomes is vital for the cells and depends both on the ESCRT machinery and lysophagy, which repair and / or remove extensively damaged compartments (*López-Jiménez et al., 2018*; *Daussy and Wodrich, 2020*). Studies in HeLa Kyoto cells have demonstrated that membrane tension plays a role in endosomal membrane remodelling and that both hyperosmotic shock and LLOMe-induced perforations, result in a decrease of membrane tension sufficient to trigger the recruitment of the ESCRT-III subunit CHMP4B (*Mercier et al., 2020*). Remarkably, we found that TrafE is recruited to endolysosomes minutes after LLOMe treatment or hyperosmotic shock (*Figures 6A, D, 7A and B*), suggesting its involvement in endolysosomal membrane remodelling and repair signalling pathways. We also demonstrate that, unlike WT *D. discoideum* cells, *trafE*-KO cells fail to restore their endolysosomal pH after LLOMe-induced sterile damage (*Figure 6G and H*). Furthermore, as observed during *M. marinum* infection, *trafE*-KO mutant cells are extremely susceptible to LLOMe compared to *atg1*-KO or WT cells (*Figure 7E*), in agreement with our hypothesis that TrafE is upstream of ESCRT and autophagic membrane repair or removal pathways. Comparison of total RNA extracted from WT and *trafE*-KO cells under basal conditions indicated misregulation of nearly all genes belonging to the autophagy and the ESCRT families (*Figure 8B*), pointing to a connection between these pathways which TrafE may link.

Therefore, we set out to discern where ESCRT and autophagy intersect and importantly how TrafE may bridge the two pathways. As part of the cell survival mechanisms, autophagy is induced upon starvation to consume cytosolic constituents and to repair or remove extensively damaged compartments. As indicated by an increase of GFP-Atg8a puncta, we demonstrated that upon starvation autophagy is initiated in both WT and *trafE*-KO cells (*Figure 9A and B*). Therefore, the absence of TrafE does not affect autophagy initiation but rather its effectiveness as revealed by accumulation of GFP-Atg8a-positive structures in *trafE*-KO cells (*Figure 9A and B*). In a similar fashion, during sterile damage, despite the modest increase of GFP-Atg8a puncta in *trafE*-KO cells compared to WT (*Figure 9C*), these autophagosome-like compartments persisted (*Figure 9D*).

The ESCRT machinery has been extensively studied for its involvement in membrane remodelling, including regulation of autophagosome maturation (*Takahashi et al., 2018*; *Zhou et al., 2019*). In the present study, we show for the first time that in *D. discoideum* cells during starvation, the ESCRT accessory protein ALIX and ESCRT-III subunit Vps32 are recruited to dextran-negative compartments (*Figure 10—figure supplement 1A, B*) in contrast to starved *atg1*-KO cells (*Figure 10—figure supplement 1C, D*). To our surprise, compared to starved WT cells, absence of TrafE lead to a strong accumulation of ALIX-GFP and GFP-Vps32 while the number of GFP-Vps4 foci remained at its basal level (*Figure 10—figure supplement 1E–G*). Importantly, we showed that in *trafE*-KO cells, LLOMe causes significant build-up of ALIX-GFP or GFP-Vps32 structures relative to WT cells (*Figure 10A, B, C and D*), without concomitant and functionally crucial GFP-Vps4 recruitment (*Figure 11A and B*, *Figure 11—figure supplement 1*). Furthermore, in complemented *trafE*-KO cells, cytosolic GFP-Vps4 was redistributed to LLOMe-damaged compartments and colocalized with mRFP-TrafE (*Figure 11C and D*). Significantly, whilst in TrafE mutant cells the number ALIX-GFP structures was abnormal in mock- and LLOMe-treated cells, the K63-polyubiquitination of damaged compartments was severely reduced (*Figure 12B, D and E*). Altogether, these results demonstrate that in *D. discoideum* cells

ESCRT and autophagy pathways intersect during endolysosomal membrane repair in which the absence of TrafE leads to striking functional disruptions of both machineries.

## Materials and methods

### *Dictyostelium discoideum* strains, culture, and plasmids

*D. discoideum* Ax2(Ka) WT and mutant strains, listed in *Supplementary file 1a*, were grown in axenic conditions at 22 °C in HL5c medium (Formedium) supplemented with 100 U/mL penicillin and 100 µg/mL streptomycin (Invitrogen). Cell lines expressing fluorescent reporters and KO cell lines were cultured in the presence of the appropriate antibiotics, hygromycin (50 µg/mL), blasticidin (5 µg/mL), or Geneticin G418 (5 µg/mL). *D. discoideum* WT, *trafE* knock-out (*trafE*-KO) or *atg1*-KO cell lines were stably transformed with GFP-TrafA, GFP-TrafB, GFP-TrafC, GFP-TrafD, GFP-TrafE, GFP-Atg8a, GFP-p62, GFP-Vps32, mCherry-Plin constructs (*Supplementary file 1c*). The *trafE*-KO was generated in Ax2(Ka) background by homologous recombination using a knock-out vector obtained following a one-step cloning in pKOSG-IBA-dicty1 (*Supplementary file 1a*) as previously described (*Wiegand et al., 2011*). The *trafE* GFP knock-in (TrafE-GFP KI) was generated in Ax2(Ka) background by homologous recombination using a knock-in vector obtained following two-step cloning in pPI183 (*Supplementary file 1a*; *Paschke et al., 2018*). WT, *trafE*-KO and *atg1*-KO knock-in *act5*::ALIX-GFP, *act5*::GFP-Vps32, *act5*::GFP-Vps4 and *act5*::GFP-Atg8a cell lines were generated following homologous recombination after one-step cloning of ALIX CDS into pDM1515 or Vps32,Vps4 and Atg8a CDSs into pDM1513 as previously described (*Paschke et al., 2018*).

### *Mycobacterium marinum* strains, culture, and plasmids

*M. marinum* WT and ΔRD1 strains (*Supplementary file 1b*) expressing GFP, mCherry or a bacterial luciferase operon (*Carroll et al., 2010*; *Andreu et al., 2010*; *Cardenal-Muñoz et al., 2017*) were cultured in shaking at 150 rpm at 32 °C in Middlebrook 7H9 (Difco) supplemented with 10% oleic acid-albumin-dextrose-catalase (OADC), 0.2% glycerol, and 0.05% Tween 80. Bacterial clumping was minimized by adding 5 mm glass beads during cultivation. Mutants and plasmid carriers were grown in medium supplemented with hygromycin (100 µg/ml) or kanamycin (25 µg/ml), as appropriate.

Infection assay. Infections were performed as previously described (*Hagedorn and Soldati, 2007*; *Arafah et al., 2013*; *Barisch et al., 2015*). *M. marinum* bacteria were spinoculated onto adherent *D. discoideum* cells and extracellular bacteria were rinsed off. The infected cells were resuspended in HL5c containing 5 U mL$^{-1}$ of penicillin and 5 µg mL$^{-1}$ of streptomycin (Invitrogen) to prevent extracellular bacterial growth. Infections were performed at a multiplicity of infection (MOI) of 10 for *M. marinum* WT in *D. discoideum* WT. MOI used for *M. marinum* ΔRD1 mutant cells was double the MOI of *M. marinum* WT. Infected cells for time points analysis were incubated at 25 °C.

### Intracellular bacteria growth assay

In this assay the bioluminescence intensity is measured in relative light units (RLU) as a readout for bacterial growth (*Sattler et al., 2007*). Measurements of luciferase-expressing *M. marinum* intracellular growth were carried out as previously described (*Arafah et al., 2013*). Three different dilutions of infected cells (0.5–2 x 10$^5$ cells/well) were plated on non-treated, white F96 MicroWell plates (Nunc) with a gas permeable moisture barrier seal (Bioconcept). Luminescence was measured at 1 hour intervals for 30–72 hours at a constant temperature of 25 °C using a Synergy Mx microplate reader (Biotek).

### Starvation assay

To induce starvation, Hl5c growth medium was removed and *D. discoideum* WT, *trafE*-KO or *atg1*-KO knock-in *act5*::ALIX-GFP, *act5*::GFP-Vps32, *act5*::GFP-Vps4 and *act5*::GFP-Atg8a cell lines were washed once with Soerensen buffer (SB, 15 mM KH$_2$PO$_4$, 2 mM Na$_2$HPO$_4$, pH 6.0) and incubated for 90 min in SorMC-Sorbitol buffer (15 mM KH$_2$PO$_4$, 2 mM Na$_2$HPO$_4$, 50 µM MgCl$_2$, 50 µM CaCl$_2$, pH 6.0, 120 mM Sorbitol) directly in 96-well black, glass-bottom plates (Perkin Elmer, MA, USA). Images from triplicates with 9 positions per well were recorded using a 60 x water immersion objective with the ImageXpress Micro XL high-content microscope (Molecular Devices). The number of ALIX-GFP, GFP-Vps32, GFP-Vps4 or GFP-Atg8a structures and foci were counted using the MetaXpress software (Molecular Devices, CA, USA), and analysed using GraphPad Prism software.

## Live fluorescence microscopy

Live microscopy was performed using inverted 3i Marianas or Nikon Ti CSU-W1 spinning disk microscopes with a 63 x glycerol / air or 100 x oil objectives. Three to 10 slices 0.5 µm apart were taken for z-stacks. For time-lapse experiments, images were acquired from non-infected or infected cells at intervals of 1 min, 5 min or at the indicated time points (hpi). Image analysis was performed using Fiji (ImageJ). *D. discoideum* cells were plated at $10^6$ /ml in 35 mm Glass Bottom Dishes (MatTek) or in 4-wells µ-slides (Ibidi). Sterile damage in *D. discoideum* was induced using 4.5–5 mM of LLOMe (Bachem) from a 10-fold concentrated solution that was added 5 min after the start of imaging. To induce a hyperosmotic shock, 2 M sorbitol was added to a final concentration of 200 mM, as described previously (*Na et al., 2007*). The lumen of endolysosomal compartments was visualized using 10–15 µg/mL of 10 kDa Alexa Fluor 647 Dextran (Invitrogen) added 3 hr prior to microscopy and washed 10–20 min before visualization of the sample. To detect neutralization of endolysosomes, 1 µM Lysosensor Green DND-189 (Invitrogen) was added 3 hr prior to microscopy and washed away 10–20 min before visualization of the sample. To distinguish live from dead cells 1 µg/mL of propidium iodide (PI) was added directly to the cell culture immediately prior to time-lapse imaging. For high-content live microscopy, *M. marinum*-infected WT and *trafE*-KO *D. discoideum* cells (as described above) were deposited in 96-well black, glass-bottom plates (Perkin Elmer, MA, USA) and left to adhere for 30 min at room temperature. Images were recorded every 1 min for 30–60 min using a ×40 or ×60 objectives with the ImageXpress Micro XL high-content microscope (Molecular Devices). The number of infected cells and extracellular bacteria or ALIX-GFP-, GFP-Atg8a-, GFP-Vps32-, and GFP-Vps32-positive structures were counted using the MetaXpress software (Molecular Devices, CA, USA), and analysed using GraphPad Prism software.

## Antibodies and immunofluorescence

*D. discoideum* cells on coverslips were rapidly fixed by immersion in –85 °C ultracold methanol as described previously (*Hagedorn et al., 2006*). The GFP fluorescent signal was enhanced using anti-GFP rabbit polyclonal antibody (MBL), anti-Ub (FK2) mouse monoclonal antibody (Enzo Life Sciences), anti-K63-linkage-specific mouse monoclonal antibody (Enzo Life Sciences). Nuclei were stained with 1 µM DAPI (Invitrogen). Cells were embedded using ProlongGold antifade (Invitrogen). As secondary antibodies we used goat anti-rabbit and anti-mouse coupled to Alexa 488 or Alexa 594 (Invitrogen). Images were acquired with a LSM700 or LSM800 microscope (Zeiss) using an oil ×63 objective. Image analysis was performed using ImageJ.

## qRT-PCR sample collection and analysis

To monitor *trafE* mRNA levels, mock- or *M. marinum*-infected WT *D. discoideum* cells were harvested, RNA was extracted, and cDNA was synthesized using the Bio-Rad iScript kit. The mean calculated threshold cycles (CT) were averaged and normalized to the CT of a reference gene with constant expression (GAPDH). The normalized CT was used to calculate the fold change using the ΔΔCT method. Relative levels of target mRNA, normalized with respect to an endogenous control (GAPDH), were expressed as 2-ΔΔCT (fold), where ΔCT = CT target gene–- CT reference gene (GAPDH), and ΔΔCT = ΔCT studied sample - ΔCT calibrator conditions.

## RNA extraction

RNA was extracted from cells using the Direct-zol RNA extraction kit (Zymo research) following the manufacturer's instructions for total RNA isolation (*Hanna et al., 2019*). To remove contaminating genomic DNA, samples were treated with 0.25 U of dNase I (Zymo) per 1 µg of RNA for 15 min at 25 °C. RNA was quantified using Qubit 4.0 (Invitrogen) and its quality was checked on the Agilent 2100 Bioanalyzer (Agilent Technologies).

## Sequencing

As previously described (*Hanna et al., 2019*), total RNAs were subjected to cDNA synthesis and NGS library construction using the Ovation Universal System (NuGEN Technologies, San Carlos, California, USA). A total of 100 ng of total DNAse I-treated RNA was used for first- and then second-strand cDNA synthesis following the manufacturer's protocol. In order to obtain comparable library size, a double bead cut strategy was applied using the 10 X genomics protocol. cDNA was recovered using

magnetic beads with two ethanol washing steps, followed by enzymatic end repair of the fragments. Next, barcoded adapters were ligated to each sample, followed by an enzymatic strand selection step and magnetic bead recovery, as above. rRNAs were targeted for depletion by the addition of custom designed oligonucleotides specific for *D. discoideum* (5 S,18S, 28 S). To amplify the libraries, 18 cycles of PCR were performed based on QC experiments carried out using RT-PCR. The quality of the libraries was monitored by TapeStation (Agilent, High Sensitivity D1000 ScreenTape, # 5067–5584). Eight-plexed samples were pooled in approximately equimolar amounts and run in 50 bp single read flow cells (Illumina, # 15022187) and run on a Hiseq 4000 (Illumina).

## RNA-seq analysis pipeline

- Alignment. RNA-seq libraries from infected and mock-treated cells taken at the indicated time points were analysed in pairwise comparisons. Fifty nt single-end reads were mapped to the *D. discoideum* genome (downloaded from dictybase) (*Fey et al., 2008*) using tophat (version 2.0.13) and bowtie2 (version 2.2.4) software. Multi hits were not allowed, by using option–-max-multi hits 1. The other parameters were set to default. The read counts per gene were generated using HTSeq software (version 0.6.1) and the GFF annotation downloaded from dictybase (February 2019). Options for htseq-count were -t exon–-stranded=yes m union.
- Analysis of differentially expressed genes. Gene counts were imported to R (version 4.0.3) filtered for remaining ribosomal gene counts (DDB_G0295651, DDB_G0295641, DDB_G0295653, DDB_G0295655, DDB_G0295647, DDB_G0295643, DDB_G0295645, DDB_G0295649, DDB_G0294034) and subsequently analysed using DESeq2 (version 1.30.1) (*Love et al., 2014*) employing the HTS filter (*Rau et al., 2013*). A binary comparison *trafE*-KO versus WT was performed, fold change was calculated using the ashr method (*Stephens, 2016*) in DESeq2, the *p* value was corrected using the Benjamini-Hochberg (*Benjamini and Hochberg, 1995*).
- Correction. The resulting false discovery rate (FDR) was used for downstream analysis. Hard thresholds to select differentially expressed genes (DE genes) were set at the absolute log2 fold change $\geq 0.585$ (equals to a fold change of 1.5) and FDR $\leq 0.05$.
- Pathway analysis of DE genes. Pathway analysis was performed using the R package clusterProfiler (*Yu et al., 2012*) and GO (downloaded from QuickGo, 2021-02-16) and KEGG (22.6.2021) annotations. For GSEA genes were ranked by false discovery rate, a group was considered significant at *p* value $\leq 0.05$, q-value $\leq 0.1$, group size between 2 and 200, using 10000 permutations.
- Heatmap of normalized counts. Gene counts were filtered for remaining ribosomal genes (DDB_G0295651, DDB_G0295641, DDB_G0295653, DDB_G0295655, DDB_G0295647, DDB_G0295643, DDB_G0295645, DDB_G0295649, DDB_G0294034) and subsequently for low expressed genes with the function filterByExpr from the edgeR package (*Robinson et al., 2009*), then normalized using the vst function from the limma package (*Ritchie et al., 2015*) and batch corrected for the effect of the experimental date using the removeBatchEffect function from the limma package. Normalized counts were subjected to the heatmap.2 function from the package gplots (3.1.1), scaling and centering the data for each gene.

## Single-cell experiments

Single-cell experiment were performed using the InfectChip device (*Delincé et al., 2016*) as described (*Mottet et al., 2021*). Briefly, the device is composed of a microfluidic chip in PDMS with a micropatterned serpentine channel, two holders (one in PMMA and one in aluminium), plastic screws and a micropatterned coverslip. Cells were infected as described above and seeded onto the coverslip. The microfluidic chip was placed onto the PMMA holder and covered with an agarose sheet prepared as explained elsewhere (*Mottet et al., 2021*). Then, the coverslip was placed onto the aluminium holder and the device was mounted and closed with the screws. A 60 mL syringe was filled with 35 mL of HL5c supplemented with 5 µg/mL of streptomycin and 5 U/mL of penicillin, and connected to the tubing of the microfluidic chip. Imaging was performed with a spinning disk confocal microscope (Intelligent Imaging Innovations Marianas SDC mounted on an inverted microscope (Leica DMIRE2)) with the following parameters: 63 X glycerol objective, transmitted light, excitation with laser lines 488 nm and 594 nm and temperature controlled at 25 °C. The loaded InfectChip device was placed on the stage and the syringe placed into a syringe pump (WPI, NE-1000) with a flow rate of 10 µL/min. Z stacks of 8 slices of 1.5 µm each were acquired every 15 min using phase contrast and every 1 h for fluorescence for 20 hr total. Image analysis for the single-cell experiments was performed as follows. Briefly, using the ImageJ software (https://imagej.net/Fiji/Downloads), cells were manually tracked,

and their time of death was recorded, and the results were presented in the form of a survival curves using the GraphPad Prism software.

## Statistics

Microscopy images were analysed using Fiji ImageJ. Experiments in *Figure 1B, C*, *Figure 2B*, *Figure 3B* and *Figure 4C, E* were quantified manually. Experiments in *Figures 6H and 7D* were quantified automatically using Fiji ImageJ Time Series Analizer V3. Experiments in *Figures 9B, C, 10C, D, 11B and 13E* were quantified using MetaXpress. Experiments in *Figures 6D, F, 7B, 12D and E* were quantified automatically using Fiji ImageJ ValelabUtils SpotCounter. Plots and statistical tests were performed using GraphPad Prism software. In all plots, standard error of the mean is shown, unless explicitly mentioned. Two-tailed t-test, ANOVA or Two-way ANOVA were used. Bonferoni multiple comparison tests were performed when necessary (n.s.: non-significant, *: p-value ≤0.05, **: p-value ≤0.01, ***: p-value ≤0.001, ****: p-value ≤0.0001).

## Acknowledgements

We acknowledge the members of the Bioimaging and ACCESS Platforms of the Faculty of Science. We thank Dr Dimitri Moreau and Dr Vincent Mercier for their help with MetaX-press and Dr Nabil Hanna for his contribution to the conception and practical realization of the RNA-seq experiment.

## Additional information

### Funding

| Funder | Grant reference number | Author |
|---|---|---|
| Schweizerischer Nationalfonds zur Förderung der Wissenschaftlichen Forschung | Project grant 310030_188813 | Lyudmil Raykov Manon Mottet Jahn Nitschke Thierry Soldati |
| Schweizerischer Nationalfonds zur Förderung der Wissenschaftlichen Forschung | Sinergia grant CRSII5_189921 | Lyudmil Raykov Manon Mottet Jahn Nitschke Thierry Soldati |

The funders had no role in study design, data collection and interpretation, or the decision to submit the work for publication.

### Author contributions

Lyudmil Raykov, Conceptualization, Resources, Data curation, Formal analysis, Validation, Investigation, Visualization, Methodology, Writing - original draft, Writing - review and editing; Manon Mottet, Formal analysis, Investigation, Visualization, Methodology; Jahn Nitschke, Data curation, Software, Formal analysis, Validation, Investigation, Visualization, Methodology, Writing - review and editing; Thierry Soldati, Conceptualization, Resources, Supervision, Funding acquisition, Investigation, Project administration, Writing - review and editing

### Author ORCIDs

Lyudmil Raykov http://orcid.org/0000-0001-7989-610X
Thierry Soldati http://orcid.org/0000-0002-2056-7931

### Decision letter and Author response

Decision letter https://doi.org/10.7554/eLife.85727.sa1
Author response https://doi.org/10.7554/eLife.85727.sa2

## Additional files

### Supplementary files

• Supplementary file 1. (a) *D. discoideum* material used in this study. The Supplementary file includes table with the *D. discoideum* strains used in this study, the overexpression plasmids and the plasmids used for generation of *trafE* knock-out and GFP knock-in. (b) *M. marinum* material used in this study. The Supplementary file includes table with the *M. marinum* strains and *M. marinum* plasmids used in this study. (c) Primers used in this study. The Supplementary file includes table with primers used to amplify *trafA*, *trafB*, *trafC* and *trafE* CDSs, primers used for knock-out, GFP knock-in generation, *act5* locus integration and screenings.

• MDAR checklist

### Data availability

All data generated or analysed during this study are included in the manuscript and supporting files; Source data files have been provided for Figures 1B, 1C, 1D, 1E, 2A, 2B, 2C, 2D, 2E, 3B, 3C, 3D, 4C, 4E, 5B, 5C, 6D, 6F, 6H, 7B, 7D, 7E, 8A, 8B, 8C, 9B, 9C, 10C, 10D, 11B, 12D, 12E and for Figure 1 - figure supplement 3A, B, C, D, 4A, B, Figure 10 - supplement 1C, 1D, 1E, 1F, 1G. The graph from Figure 1 - figure supplement 2 was generated by analysis of publicly available data (*Hanna et al., 2019*). The MCV proteomics information was obtained from a publicly available dataset (*Guého et al., 2019*).

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

## Apppendix 1

**Appendix 1—key resources table**

| Reagent type (species) or resource | Designation | Source or reference | Identifiers | Additional information |
|---|---|---|---|---|
| Gene (*Dictiostelium discoideum*) | *trafA* | Dictybase | DDB_G0272454 | |
| Gene (*Dictiostelium discoideum*) | *trafB* | Dictybase | DDB_G0285149 | |
| Gene (*Dictiostelium discoideum*) | *trafC* | Dictybase | DDB_G0290883 | |
| Gene (*Dictiostelium discoideum*) | *trafD* | Dictybase | DDB_G0290961 | |
| Gene (*Dictiostelium discoideum*) | *trafE* | Dictybase | DDB_G0290971 | |
| Cell line (*Dictiostelium discoideum*) | Ax2(Ka) | Dictybase | WT, parental strain of the *trafE* KO, *atg1* KO and GFP-trafE KI | |
| Cell line (*M. marinum*) | M | L.Ramakrishnan (University of Cambridge) | WT | |
| Cell line (*Dictiostelium discoideum*) | Ax2(Ka) | This paper | *trafE* KO | |
| Cell line (*Dictiostelium discoideum*) | Ax2(Ka) | This paper | *atg1* KO | |
| Cell line (*Dictiostelium discoideum*) | Ax2(Ka) | This paper | GFP-trafE KI | Cell line maintained in T. Soldati lab |
| Cell line (*Dictiostelium discoideum*) | Ax2(Ka) | This paper | WT; *act5*::Atg8a | Cell line maintained in T. Soldati lab |
| Cell line (*Dictiostelium discoideum*) | Ax2(Ka) | This paper | WT; *act5*::ALIX | Cell line maintained in T. Soldati lab |
| Cell line (*Dictiostelium discoideum*) | Ax2(Ka) | This paper | WT; *act5*::Vps32 | Cell line maintained in T. Soldati lab |
| Cell line (*Dictiostelium discoideum*) | Ax2(Ka) | This paper | WT; *act5*::Vps4 | Cell line maintained in T. Soldati lab |
| Cell line (*Dictiostelium discoideum*) | Ax2(Ka) | This paper | trafE KO; *act5*::Atg8a | Cell line maintained in T. Soldati lab |
| Cell line (*Dictiostelium discoideum*) | Ax2(Ka) | This paper | trafE KO; *act5*::ALIX | Cell line maintained in T. Soldati lab |
| Cell line (*Dictiostelium discoideum*) | Ax2(Ka) | This paper | trafE KO; *act5*::Vps32 | Cell line maintained in T. Soldati lab |
| Cell line (*Dictiostelium discoideum*) | Ax2(Ka) | This paper | trafE KO; *act5*::Vps4 | Cell line maintained in T. Soldati lab |

*Appendix 1 Continued on next page*

*Appendix 1 Continued*

| Reagent type (species) or resource | Designation | Source or reference | Identifiers | Additional information |
|---|---|---|---|---|
| Cell line (*M. marinum*) | ΔRD1 | L.Ramakrishnan (University of Cambridge) *Volkman et al., 2004* | RD1 locus ablation mutant | |
| Antibody | anti-GFP (rabbit polyclonal) | MBL | 598 | IF(1:1000), WB (1:1000) |
| Antibody | anti-Ub (FK2) (Mouse monoclonal) | Enzo Life Sciences | BML-PW8810 | IF(1:1000) |
| Antibody | anti-K63-linkage-specific (Mouse monoclonal) | Enzo Life Sciences | HWA4C4 | IF(1:50) |
| Recombinant DNA reagent | pDM317 (plasmid) | *Veltman et al., 2009* | Dictybase | GFP (N-terminal on backbone) |
| Recombinant DNA reagent | pDM1513 (plasmid) | *Paschke et al., 2018* | 108998 | GFP (N-terminal on backbone) |
| Recombinant DNA reagent | pDM1515 (plasmid) | *Paschke et al., 2018* | 109000 | GFP (C-terminal on backbone) |
| Recombinant DNA reagent | pDM318 (plasmid) | *Veltman et al., 2009* | Dictybase | dsRed (N-terminal on backbone) |
| Sequence-based reagent | LR32F | This paper | PCR primers; trafE 5' | CAGGATCCAAAATGACAGTAAAATATTCAATTAATG |
| Sequence-based reagent | LR32R | This paper | PCR primers; trafE 3' | CAACTAGTTGGTAAAACTTGAATTCTAAG |
| Sequence-based reagent | LR63F | This paper | PCR primers; trafE; qPCR | GAGTCTTGTAAAAAATCATTCCCAAG |
| Sequence-based reagent | LR63R | This paper | PCR primers; trafE; qPCR | GTTGGTTATTTATAACTTTGTCCATC |
| Sequence-based reagent | LR7F | This paper | PCR primers; trafA 5' | CAGGATCCAAAATGGATATTTCTCAAATCC |
| Sequence-based reagent | LR7R | This paper | PCR primers; trafA 3' | CAACTAGTATGTTTATCACATTGAGAC |
| Sequence-based reagent | LR8F | This paper | PCR primers; trafB 5' | CAGGATCCAAAATGACAGAGTTTAAAATTAG |
| Sequence-based reagent | LR8R | This paper | PCR primers; trafB 3' | CAACTAGTTTTAGTAGTTAAAGGATC |
| Sequence-based reagent | LR9/10 F | This paper | PCR primers; trafC 5' | CAGGATCCAAAATGTCAATTGATATAAAATTTAC |
| Sequence-based reagent | LR9R | This paper | PCR primers; trafC 3' | CAACTAGTAGACTCCAATGGTTCATATTC |
| Sequence-based reagent | LR9/10 F | This paper | PCR primers; trafD 5' | CAGGATCCAAAATGTCAATTGATATAAAATTTAC |
| Sequence-based reagent | LR10R | This paper | PCR primers; trafD 3' | CAACTAGTAGACTCCAATGGTTCATATTC |

*Appendix 1 Continued on next page*

*Appendix 1 Continued*

| Reagent type (species) or resource | Designation | Source or reference | Identifiers | Additional information |
|---|---|---|---|---|
| Commercial kit | RNA extraction kit Direct-zol | Zymo research | R2062 | |
| Other | Lysosensor Green DND-189 | Invitrogen | L7535 | Fluorescent Dye;1 µM |
| Other | 10 kDa Alexa Fluor 647 Dextran | Invitrogen | D22914 | Alexa fluorophore labelled dextran;10 µg/mL |
| Other | DAPI stain | Invitrogen | D1306 | 1:50 |
| Other | PI stain | ThermoFisher | R37108 | 1 µg/mL |
| Chemical compound | LLOMe | Bachem | 16689-14-8 | 4.5–5 mM |

