## [Editor Report]

This study presents important findings on the mechanism as to how *Mycobacterium*-containing vacuoles are recognized by host cell factors and subjected to membrane repairment or autophagic degradation using *Dictyostelium discoideum* as a useful model. The evidence for the role of TrafE in damaged membrane repair and xenophagy induction is convincing. This work will be of interest to cell biologists and microbiologists.

---

## [Decision Letter]

**Decision letter after peer review:**

Thank you for submitting your article "A TRAF-like E3 ubiquitin ligase TrafE coordinates endolysosomal damage response and cell-autonomous immunity to Mycobacterium marinum." for consideration by *eLife*. Your article has been reviewed by 3 peer reviewers, including Noboru Mizushima as the Reviewing Editor and Reviewer #1, and the evaluation has been overseen by David Ron as the Senior Editor.

Essential revisions:

1) While the studies on the involvement of TrafE in damaged membrane repair and xenophagy induction are mostly convincing, the studies on phagophore closure are indirect and too preliminary. If the authors want to retain this part, they need to address the questions and concerns raised by Reviewer #1 and Reviewer #3 (comment 1). Alternatively, it is also suggested that the authors significantly tone down or even remove the section on autophagosome closure from this paper and focus on membrane repair and xenophagy induction (as this manuscript already has 13 figures). It would not reduce this paper's novelty and rather make it more convincing (and save time).

2) The authors overinterpret some of the data (e.g., see Reviewer #3-Comments #9, 10, 11, 13, 14, and 15). Some of the statements are also confusing (Reviewer #3-Comments #2, 4, 6, and 13). Please rewrite the text more carefully not to avoid confusion or provide experimental evidence.

3) Page 4, line 13: Provide sequence alignment of Dd TRAFs, indicate 'stereotypical' domains, build a phylogenetic tree, and reveal the closest homologue to TrafE (Reviewer #2-Comment 3).

4) Figure S3: Please add TrafE KO as a control (Reviewer #3-Comment 7).

5) Figure 2: Please provide representative images for Plin recruitment to bacteria (Reviewer #2-Comment 8).

6) Figure 11-13: The authors should monitor the recruitment of endogenous or non-tagged ESCRT components in TrafE KO cells (Reviewer #2-comment 2).

The other comments from the three reviewers are useful and would definitely improve the manuscript.

*Reviewer #1 (Recommendations for the authors):*

The part of phagophore closure is too preliminary. I would suggest that the authors should significantly weaken the conclusion or move this part to discussion. If the authors want to retain this part, they need to address the following questions.

– Is TrafE recruited to autophagosomes before sealing? (LC3-positive and dextran-negative structures).

– Are autophagosomes not closed in trafEKO cells? The method used by Takahashi et al. 2018 may be useful.

– Is starvation-induced autophagic flux indeed reduced in trafEKO cells (even though the development of trafEKO is normal)?

– Is the recruitment of upstream ATGs (e.g., ULK1) normal in trafEKO cells (Figure 10)?

– Are the behaviors of ATG8 and ALIX similar in vps4KO cells (or VPS4 dominant negative-expressing cells) and trafEKO cells?

*Reviewer #2 (Recommendations for the authors):*

– The author proposed the model that TrafE plays a role in phagophore sealing during autophagosome maturation, as evidenced by the observation of increased Atg8 positive puncta in TrafE-deficient cells under starvation conditions. It appears that there is insufficient data to formulate such a model. Typically, the increase in Atg8 positive puncta can be attributed to two factors: the upregulation of autophagy induction and the accumulation of autophagosomes due to the inhibition of autolysosome formation. To further clarify these points, the author needs to investigate the flow of autophagosome formation and assess any changes in cargo protein levels.

– In TrafE knockout cells, the recruitment of Vps4 was not detected, while Vps32 and ALIX were found to accumulate at sites of membrane damage. The author evaluated the recruitment of Vps32, ALIX and Vps4 in TrafE knockout cells through the use of GFP fusion proteins. Previous reports have demonstrated that the overexpression of large protein (GFP and RFP)-fused ESCRT proteins, particularly ESCRT-III proteins, can disrupt the entire ESCRT pathway (e.g., von Schwedler et al., 2003, Cell, 114:701). These results may not accurately reflect the normal behavior of Vps32, ALIX, and/or Vps4. It is crucial to monitor endogenous or non-tagged proteins in these experiments.

*Reviewer #3 (Recommendations for the authors):*

The authors investigate the response of *Dictyostelium* to endolysosomal damage upon infection with Mycobacteria. They report the identification of TrafE as a component of the ELDR pathway required for the restriction of bacterial growth.

The manuscript has lots of interesting new data and the identification of a new gene involved in the detection/clearance of damaged endomembranes is clearly worth reporting. However, in its current form, the manuscript is not suitable for immediate publication – the text is way too long, data are not always presented in a logical order, the number of main figures (n=13!) is unlikely to be publishable, and, most importantly, the manuscript contains certain (sloppily written) conclusions that are merely consistent with the data but that by no means have been demonstrated with reasonable certainty. The comments/questions below are meant to provide the authors with examples to illustrate my points.

1) What is the nature of the TrafE structures on damaged MCVs close to bacterial poles? Are they directly associated with damaged membranes or do they form a protein aggregate merely initiated by membrane damage?

2) Abstract: Rather confusing. ELDR is introduced as the 'upstream endolysosomal damage response (ELDR) that senses damage … and is required for the recruitment of ELDR components.'

3) Page 4, line 13: provide sequence alignment of Dd TRAFs, indicate 'stereotypical' domains, build a phylogenetic tree, reveal the closest homologue to TrafE.

4) Page 4 -18: 'overexpressed' is the wrong term to describe the upregulation of a gene during infection, particularly since later the authors exogenously overexpress the same gene.

5) Page 4-32: quantify and compare recruitment of at least TrafC, TrafD, and TrafE. Specify aa differences between TrafC and TrafD, point out domains, and are the differences predicted to be surface exposed?

6) Page 4-35: Again, I would like to encourage the authors to edit their text for readability. Why would they initially mention 'overexpression' (i.e. upregulation) of TrafE in line 19, then come back to it in line 36.

7) Page 5-12: authors claim that C-terminal fusion of TrafE:GFP is functional because they don't see a growth phenotype in FigS3. The argument is flawed, however, as they haven't shown that cells deficient in TrafE would have a phenotype. (Half a page later, in line 22, the authors then generate the badly needed trafE ko line. Clearly, the manuscript flow is not good. Please re-organize.)

8) Page 6-1: please provide representa>.

9) Page 6-27: The authors claim that lack of TrafE '… strongly affects targeting of bacteria by the autophagy machinery. This claim comes completely out of the blue and seems unfunded. What is the evidence that TrafE affects anti-bacterial autophagy?

10) Page 6-28: '… ATG1 and TrafE work in the same pathway'. There is little evidence in the manuscript to support this bold claim. If the authors want to keep the claim, an epistatic experiment would be needed.

11) Page 7-14: The authors claim that TrafE is upstream of '…subsequent restriction of bacteria proliferation by autophagy.' The authors provide no evidence for their claim.

12) Page 7-32: A cartoon with the domain structure of TrafE and the truncations generated would greatly benefit the reader.

13) Page 8-23: '… TrafE is an early sensor of damage and an upstream effector of the ELDR'. I would encourage the authors to write more precisely. Their sentence suggests that the authors think there are early and late sensors?! And why would TrafE be a sensor and an effector? I just don't understand their ELDR nomenclature, similar to my comments about ELDR in the abstract…

14) Page 9-31: The authors claim that the TrafE phenotype is a '…cumulative consequence of non-functional autophagy- and ESCRT-dependent membrane repair.' This is speculation – an attractive hypothesis maybe – but the authors need to perform actual experiments to prove their point. An epistatic analysis would nail it.

15) Page 10-32: "…our data indicate that TrafE acts downstream of ATG1 in autophagy…" While this is a reasonable hypothesis, the authors do not provide definitive data for TrafE acting downstream of ATG1.

16) Legend page 27-3: Dd were not infected for x hours etc but assayed at x hpi.

17) Figure 1D, E: y-axis labeled with "(FC)". Pls explain

18) Figure S2: Show all data at the same magnification (as in A). Did TrafD in FigS2D change its localization upon infection and become clustered? If so, pls comment in the text.

19) Fig3D: The authors claim lack of cell death upon infection with RD1 deficient bacteria but lacking a positive control for cell death (i.e. wt bacteria into trafE ko)

20) Figure 4: There is very little overlap between TrafE and K63 Ub. Can the authors comment, please? Is this a representative image regarding the lack of colocalization between TrafE and K63?

21) Figures throughout:

Add scale bars.

Reduce number of figures (currently n=13!) by building composite figures.

---

## [Author Response]

Essential revisions:1) While the studies on the involvement of TrafE in damaged membrane repair and xenophagy induction are mostly convincing, the studies on phagophore closure are indirect and too preliminary. If the authors want to retain this part, they need to address the questions and concerns raised by Reviewer #1 and Reviewer #3 (comment 1). Alternatively, it is also suggested that the authors significantly tone down or even remove the section on autophagosome closure from this paper and focus on membrane repair and xenophagy induction (as this manuscript already has 13 figures). It would not reduce this paper's novelty and rather make it more convincing (and save time).

We thank the reviewers for the overall positive evaluation of the study. The main focus of our paper is to share findings on TrafE involvement in membrane repair. Therefore, we do agree that claiming a role of TrafE in autophagosome closure is rather speculative and that omitting the claim strengthens our manuscript.

2) The authors overinterpret some of the data (e.g., see Reviewer #3-Comments #9, 10, 11, 13, 14, and 15). Some of the statements are also confusing (Reviewer #3-Comments #2, 4, 6, and 13). Please rewrite the text more carefully not to avoid confusion or provide experimental evidence.

We apologise for the lack of clarity. We have extensively modified the text to more rigorously match data and interpretations, and to enhance clarity.

3) Page 4, line 13: Provide sequence alignment of Dd TRAFs, indicate 'stereotypical' domains, build a phylogenetic tree, and reveal the closest homologue to TrafE (Reviewer #2-Comment 3).

We thank the reviewers for this request. The domain architecture and extensive phylogenetic relationship analysis of *D. discoideum* TRAF-like proteins and TRAFs from different phyla was already published (Dunn et al., 2018). Nevertheless, we think that showing human TRAF6 and *D. discoideum* TRAF homologs in a multiple sequence alignment, the phylogenetic tree, percent-identity matrix and AlphaFold predictions of protein structure (Figure 1 —figure supplement 1A, B, C, D, E) significantly improves our manuscript.

4) Figure S3: Please add TrafE KO as a control (Reviewer #3-Comment 7).

We agree with the reviewer that the *trafE*-KO is an important control in this experiment, it was added in Figure 1 —figure supplement 4B.

5) Figure 2: Please provide representative images for Plin recruitment to bacteria (Reviewer #2-Comment 8).

We thank the reviewers for this request. We now provide representative images in Figure 2 —figure supplement 1B.

6) Figure 11-13: The authors should monitor the recruitment of endogenous or non-tagged ESCRT components in TrafE KO cells (Reviewer #2 comment 2).

We agree with the reviewer #2 – comment 2. Unfortunately, commercial antibodies against *D. discoideum* ALIX, Vps32 or Vps4 are not available, and antibodies directed to animal proteins do not cross-react. In addition, our attempts to rise such antibodies on multiple occasions were unsuccessful. However, to respond to the reviewer’s concern, we addressed whether ALIX, Vps32 or Vps4 GFP-fusions and / or overexpression may disrupt the basic ESCRT functions. As previously reported, a functional ESCRT machinery is required for the accomplishment of the developmental cycle of *D. discoideum* (Mattei et al., 2005). Therefore, we starved WT *D. discoideum* cells, or stable cell lines expressing ALIX-GFP, GFP-Vps32 or GFP-Vps4 for 24 hours and monitored that all strains complete their developmental cycle and formed spore-containing sori (see Author response image 1). In addition, previous studies have found that in human cells, overexpression of Vps4 may also be associated with cytokinesis defects (Morita et al., 2007). This ESCRT function in cytokinesis was tested as follow. *D. discoideum* cells need to be attached to a substrate in order to efficiently complete cytokinesis. Cells grown in shaking suspension cannot attach, and without traction-aided cytokinesis, often carry out karyokinesis without subsequent cytokinesis and thus tend to become multinucleated. Therefore, we grew in shaking conditions WT *D. discoideum* cells, and cells overexpressing ALIX, Vps32 or Vps4 GFP-fusions. At 72 hours, cells were transferred to tissue culture dishes, where they adhere. They were fixed immediately or after 6 hours and imaged by high-content (HC) confocal microscopy to quantify the number of nuclei per cell using the MetaXpress HC software. As expected, we observed that immediately after growth in suspension most cells had between 2 and 4 nuclei (see Author response image 2), however after 6 hours under adherent conditions, the majority of the cells had divided and reached between 1 and 2 nuclei, suggesting that overexpression of ESCRT subunits did not impact cell division. We estimate that these control experiments are important but are presented here only for Reviewers’ sake, and were not included in the revised manuscript.

**Author response image 1. sa2fig1:** 

The other comments from the three reviewers are useful and would definitely improve the manuscript.

We have taken into account all the suggestions and criticisms, see below.

Reviewer #1 (Recommendations for the authors):The part of phagophore closure is too preliminary. I would suggest that the authors should significantly weaken the conclusion or move this part to discussion. If the authors want to retain this part, they need to address the following questions.– Is TrafE recruited to autophagosomes before sealing? (LC3-positive and dextran-negative structures).– Are autophagosomes not closed in trafEKO cells? The method used by Takahashi et al. 2018 may be useful.– Is starvation-induced autophagic flux indeed reduced in trafEKO cells (even though the development of trafEKO is normal)?– Is the recruitment of upstream ATGs (e.g., ULK1) normal in trafEKO cells (Figure 10)?– Are the behaviors of ATG8 and ALIX similar in vps4KO cells (or VPS4 dominant negative-expressing cells) and trafEKO cells?

As already discussed above, the main focus of our study and manuscript is to share our finding on TrafE involvement in membrane damage repair. Therefore, we do agree that claiming a role for TrafE in autophagosome closure is rather speculative at this point and that by omitting the claim we strengthen our manuscript. However, we have addressed all other issues mentioned by the reviewer.

Reviewer #2 (Recommendations for the authors):– The author proposed the model that TrafE plays a role in phagophore sealing during autophagosome maturation, as evidenced by the observation of increased Atg8 positive puncta in TrafE-deficient cells under starvation conditions. It appears that there is insufficient data to formulate such a model. Typically, the increase in Atg8 positive puncta can be attributed to two factors: the upregulation of autophagy induction and the accumulation of autophagosomes due to the inhibition of autolysosome formation. To further clarify these points, the author needs to investigate the flow of autophagosome formation and assess any changes in cargo protein levels.

As already discussed above, the main focus of our study and manuscript is to share our finding on TrafE involvement in membrane damage repair. Therefore, we do agree that claiming a role for TrafE in autophagosome closure is rather speculative at this point and that by omitting the claim we strengthen our manuscript. However, we have addressed all other issues mentioned by the reviewer.

– In TrafE knockout cells, the recruitment of Vps4 was not detected, while Vps32 and ALIX were found to accumulate at sites of membrane damage. The author evaluated the recruitment of Vps32, ALIX and Vps4 in TrafE knockout cells through the use of GFP fusion proteins. Previous reports have demonstrated that the overexpression of large protein (GFP and RFP)-fused ESCRT proteins, particularly ESCRT-III proteins, can disrupt the entire ESCRT pathway (e.g., von Schwedler et al., 2003, Cell, 114:701). These results may not accurately reflect the normal behavior of Vps32, ALIX, and/or Vps4. It is crucial to monitor endogenous or non-tagged proteins in these experiments.

We thank the reviewer for these comments. We offer the same answer as for the reviewer 1 – comment 6: Please, see pages 2 and 3 for the full argumentation.

Reviewer #3 (Recommendations for the authors):The authors investigate the response of *Dictyostelium* to endolysosomal damage upon infection with Mycobacteria. They report the identification of TrafE as a component of the ELDR pathway required for the restriction of bacterial growth.The manuscript has lots of interesting new data and the identification of a new gene involved in the detection/clearance of damaged endomembranes is clearly worth reporting. However, in its current form, the manuscript is not suitable for immediate publication – the text is way too long, data are not always presented in a logical order, the number of main figures (n=13!) is unlikely to be publishable, and, most importantly, the manuscript contains certain (sloppily written) conclusions that are merely consistent with the data but that by no means have been demonstrated with reasonable certainty. The comments/questions below are meant to provide the authors with examples to illustrate my points.

We thank the reviewer for the very positive acknowledgement of the novelty and significance of the study and manuscript content. We apologise if the form did not match expectations. We have merged Figure 6 and 7, but would like to respectfully argue that our preference goes to “one figure one message” instead of figures composed of a myriad stamp-size panels. We also expect that, because articles nowadays appear only online, the number of figures is a minor issue.

1) What is the nature of the TrafE structures on damaged MCVs close to bacterial poles? Are they directly associated with damaged membranes or do they form a protein aggregate merely initiated by membrane damage?2) Abstract: Rather confusing. ELDR is introduced as the 'upstream endolysosomal damage response (ELDR) that senses damage … and is required for the recruitment of ELDR components.'

We thank the reviewer for this comment. We now avoid mentioning ELDR and have improved the language overall to avoid possible confusions or over-interpretations.

3) Page 4, line 13: provide sequence alignment of Dd TRAFs, indicate 'stereotypical' domains, build a phylogenetic tree, reveal the closest homologue to TrafE.

We thank the reviewer for this request. As mentioned above, (page 1, essential comment N° 3) we have now introduced this important information in Figure 1 —figure supplement 1A, B, C, D, E.

4) Page 4 -18: 'overexpressed' is the wrong term to describe the upregulation of a gene during infection, particularly since later the authors exogenously overexpress the same gene.

We apologise for this confusion. We have modified the text accordingly.

5) Page 4-32: quantify and compare recruitment of at least TrafC, TrafD, and TrafE. Specify aa differences between TrafC and TrafD, point out domains, and are the differences predicted to be surface exposed?

We thank the reviewer for this comment. The requested quantifications are now introduced in Figure 1 —figure supplement 3A, B, C, D. The differences between the TRAFs and the coordinates of their predicted domains are presented in Figure 1 —figure supplement 1C and Figure 5 —figure supplement 1A, B,

6) Page 4-35: Again, I would like to encourage the authors to edit their text for readability. Why would they initially mention 'overexpression' (i.e. upregulation) of TrafE in line 19, then come back to it in line 36.

We thank the reviewer for this comment, and we apologise for the inconsistency. We have introduced the necessary modifications in the text.

7) Page 5-12: authors claim that C-terminal fusion of TrafE:GFP is functional because they don't see a growth phenotype in FigS3. The argument is flawed, however, as they haven't shown that cells deficient in TrafE would have a phenotype. (Half a page later, in line 22, the authors then generate the badly needed trafE ko line. Clearly, the manuscript flow is not good. Please re-organize.)

We agree with the reviewer that this *trafE*-KO is an important control in this experiment. It has been added in Figure 1 —figure supplement 4B.

8) Page 6-1: please provide representa>.

We guessed that the reviewer was asking us to add representative images, which are now provided in Figure 2 —figure supplement 1B.

9) Page 6-27: The authors claim that lack of TrafE '… strongly affects targeting of bacteria by the autophagy machinery. This claim comes completely out of the blue and seems unfunded. What is the evidence that TrafE affects anti-bacterial autophagy?

We apologise for the lack of clarity. From previous studies, we know that in *D. discoideum* xenophagy is the main pathway for *M. marinum* restriction and that it is MCV membrane damage-dependent (Cardenal-Muñoz et al., 2017; López-Jiménez et al., 2018). Cytosolic bacteria are captured by the xenophagy machinery and restricted, whereas the *M. marinum* RD1 mutant that does not escape its MCV to the cytosol is unaffected by the non-functional xenophagy machinery in *atg1*-KO cells. (Cardenal-Muñoz et al., 2017; López-Jiménez et al., 2018). Therefore, our conclusions are based on the observation that the unrestricted *M. marinum* intracellular growth monitored in *trafE*-KO cells (Figure 2A, 3C) phenocopies what is seen in the xenophagy-deficient *atg1*-KO. In addition, among the dozens of KO strains in the cell-autonomous defence pathways generated in the lab, this extreme phenotype has been observed only in the *trafE*- and *atg1*-KO strains.

10) Page 6-28: '… ATG1 and TrafE work in the same pathway'. There is little evidence in the manuscript to support this bold claim. If the authors want to keep the claim, an epistatic experiment would be needed.

We apologise for this overstatement. As argued above, our result clearly indicate that *M. marinum* intracellular growth in *trafE*-KO cells is as unrestricted as in the *atg1*-KO mutant, revealing that they are both involved in xenophagy. But we do agree that the claim that Atg1 and TrafE unequivocally work in the same pathway is at this point premature. We have rephrased this in the text.

11) Page 7-14: The authors claim that TrafE is upstream of '…subsequent restriction of bacteria proliferation by autophagy.' The authors provide no evidence for their claim.

We apologise for the lack of clarity. We observed that TrafE is not recruited to *M. marinum* RD1-containing MCVs, namely in absence of MCV damage. In addition, the intracellular growth of *M. marinum* RD1 mutants is unaffected by the presence or absence of TrafE, leading to the conclusion that xenophagy-mediated restriction depends on the TrafE-initiated cascade at the site of MCV damage.

12) Page 7-32: A cartoon with the domain structure of TrafE and the truncations generated would greatly benefit the reader.

We thank the reviewer for this request. We provided representative cartoons in Figure 5A, Figure 5 —figure supplement 5A, B.

13) Page 8-23: '… TrafE is an early sensor of damage and an upstream effector of the ELDR'. I would encourage the authors to write more precisely. Their sentence suggests that the authors think there are early and late sensors?! And why would TrafE be a sensor and an effector? I just don't understand their ELDR nomenclature, similar to my comments about ELDR in the abstract…

We thank the reviewer for this comment, and we apologise for the lack of clarity. We have introduced the necessary modifications in the text.

14) Page 9-31: The authors claim that the TrafE phenotype is a '…cumulative consequence of non-functional autophagy- and ESCRT-dependent membrane repair.' This is speculation – an attractive hypothesis maybe – but the authors need to perform actual experiments to prove their point. An epistatic analysis would nail it.

We again apologise for the lack of clarity. As presented above, our results indicate that TrafE, like Atg1, is involved in the xenophagy-mediated restriction pathway for cytosolic bacteria. In addition, absence of TrafE has a more severe consequence for cells under infection- and chemical-induced damage, namely cells die. Therefore, we did not claim, but respectfully suggested (see “clean copy” of the manuscript, page 9 line 29) that in addition to initiate autophagy/xenophagy, TrafE also stimulates ESCRT-dependent endolysosome and MCV repair.

15) Page 10-32: "…our data indicate that TrafE acts downstream of ATG1 in autophagy…" While this is a reasonable hypothesis, the authors do not provide definitive data for TrafE acting downstream of ATG1.

We thank the reviewer for asking for clarification. We have pointed out in Essential Revisions – point #2 that we know from previous studies that starvation triggers autophagy and an increase of LC3/Atg8 puncta (Cardenal-Muñoz et al., 2017; López-Jiménez et al., 2018). Upon starvation, GFP-Atg8a foci increase in *trafE* KO cells, showing that autophagy is induced. However, in *trafE*-KO mutant cells we observed a stronger increase and likely abnormal accumulation of GFP-Atg8a puncta compared to WT cells, allowing us to “…suggest that the *D. discoideum* E3-ligase TrafE acts downstream of Atg1 in the autophagy pathway.” (see “clean copy” of the manuscript, page 10 line 30). Therefore, we decided to keep this softly phrased speculation in the text.

16) Legend page 27-3: Dd were not infected for x hours etc but assayed at x hpi.

We apologise for the lack of clarity. We have introduced the necessary modifications in the text.

17) Figure 1D, E: y-axis labeled with "(FC)". Pls explain

We apologise for the lack of clarity. We have introduced the necessary modifications in the text.

18) Figure S2: Show all data at the same magnification (as in A). Did TrafD in FigS2D change its localization upon infection and become clustered? If so, pls comment in the text.

We thank the reviewer for bringing this to our attention. All figures in Figure 1 —figure supplement 3A, B, C, D are now shown at the same magnification. TrafD appears to redistribute to small foci and structures with diverse shapes and sizes under both mock or infection conditions. However, during infection with *M. marinum* we have not been able to detect a pronounced colocalization with MCVs, which was the “behaviour” for which we monitored the *D. discoideum* TRAF homologs.

19) Fig3D: The authors claim lack of cell death upon infection with RD1 deficient bacteria but lacking a positive control for cell death (i.e. wt bacteria into trafE ko)

We thank the reviewer for giving us the opportunity to clarify. We now emphasize in the text that the results presented in Figure 3D were acquired in the same experiment as those shown in Figure 2E.

20) Figure 4: There is very little overlap between TrafE and K63 Ub. Can the authors comment, please? Is this a representative image regarding the lack of colocalization between TrafE and K63?

We thank the reviewer for this comment. We do detect some overlap between TrafE and K63 Ub (Figure 4A) immediately next to regions of the membrane that are probably disrupted and are devoid of both, TrafE and K63 Ub. Please, note that mechanistically, one does not expect a major degree of colocalization between an enzyme and the product of its activity.

21) Figures throughout:Add scale bars.

We thank the reviewer for this comment. We have added scale bars wherever necessary.

Reduce number of figures (currently n=13!) by building composite figures.

We have merged Figure 6 and 7, but for the rest, we decided to keep the original structure for readability and clarity.

References

Cardenal-Muñoz, E., Arafah, S., López-Jiménez, A. T., Kicka, S., Falaise, A., Bach, F., Schaad, O., King, J. S., Hagedorn, M. and Soldati, T. 2017. *Mycobacterium marinum* antagonistically induces an autophagic response while repressing the autophagic flux in a TORC1- and ESX-1-dependent manner. PLoS Pathogens, 13, e1006344.

Dunn, J. D., Bosmani, C., Barisch, C., Raykov, L., Lefrançois, L. H., Cardenal-Muñoz, E., López-Jiménez, A. T. and Soldati, T. 2018. Eat prey, live: *Dictyostelium discoideum* as a model for cell-autonomous defenses. Frontiers in Immunology, 8, 1906-1906.

López-Jiménez, A. T., Cardenal-Muñoz, E., Leuba, F., Gerstenmaier, L., Barisch, C., Hagedorn, M., King, J. S. and Soldati, T. 2018. The ESCRT and autophagy machineries cooperate to repair ESX-1-dependent damage at the Mycobacterium-containing vacuole but have opposite impact on containing the infection. PLoS Pathog, 14, e1007501.

Mattei, S., Ryves, W. J., Blot, B., Sadoul, R., Harwood, A. J., Satre, M., Klein, G. and Aubry, L. 2005. Dd-Alix, a conserved endosome-associated protein, controls *Dictyostelium* development. Developmental Biology, 279, 99-113.

Morita, E., Sandrin, V., Chung, H. Y., Morham, S. G., Gygi, S. P., Rodesch, C. K. and Sundquist, W. I. 2007. Human ESCRT and ALIX proteins interact with proteins of the midbody and function in cytokinesis. Embo j, 26, 4215-27.